# Nanoscale regulation of Ca$^{2+}$ dependent phase transitions and real-time dynamics of SAP97/hDLG

Premchand Rajeev [1], Nivedita Singh[1], Adel Kechkar[2], Corey Butler [3], Narendrakumar Ramanan [1], Jean-Baptiste Sibarita [3], Mini Jose[1] & Deepak Nair [1✉]

Synapse associated protein-97/Human Disk Large (SAP97/hDLG) is a conserved, alternatively spliced, modular, scaffolding protein critical in regulating the molecular organization of cell-cell junctions in vertebrates. We confirm that the molecular determinants of first order phase transition of SAP97/hDLG is controlled by morpho-functional changes in its nanoscale organization. Furthermore, the nanoscale molecular signatures of these signalling islands and phase transitions are altered in response to changes in cytosolic Ca$^{2+}$. Additionally, exchange kinetics of alternatively spliced isoforms of the intrinsically disordered region in SAP97/hDLG C-terminus shows differential sensitivities to Ca$^{2+}$ bound Calmodulin, affirming that the molecular signatures of local phase transitions of SAP97/hDLG depends on their nanoscale heterogeneity and compositionality of isoforms.

[1] Centre for Neuroscience, Indian Institute of Science, Bangalore, Karnataka 560012, India. [2] Ecole Nationale Supérieure de Biotechnologie, Constantine, Algeria. [3] Université Bordeaux, CNRS, Interdisciplinary Institute for Neuroscience, IINS, UMR 5297, 33000 Bordeaux, France. ✉email: deepak@iisc.ac.in

Membrane-associated guanylate kinases (MAGUKs) are multidomain scaffolding molecules that organize near membrane signaling complexes at the postsynaptic density of neurochemical synapses and other cell junctions[1,2]. Discs large (DLG) subfamily of MAGUKs is considered pivotal in the molecular organization of cell adhesion complexes in neurons and epithelial cells. Members of the DLG family comprises of SAP97/hDLG, SAP90/PSD95, SAP102/NE-DLG, and PSD93/Chapsyn 110[3]. DLG family of molecules have a unique modular organization and is composed of multiple protein–protein interaction domains. All the DLG family of molecules, characterized to date, contain three PDZ (PSD95/DLG/ZO1) domains, an Src homology three domain (SH3) and a guanylate kinase domain[1,3]. Best characterized of these domains are the PDZ domains[1,4], which binds with high affinity to the carboxyl terminal peptide motifs of several proteins, notably GluN2 subunits of NMDA receptors[5,6], GluA1 subunit of AMPA receptors[7,8] and the voltage-gated inwardly rectifying $K^+$ channels[9,10]. The guanylate kinase like (GUK) domain of MAGUKs lacks vital amino acid residues required for ATP/GMP binding[1,11]. Previous reports confirm that instead of an enzymatic role, this domain is modified to be part of multiple protein–protein interactions[4,12,13]. Accordingly, interactions have been mapped to this region including that of multidomain proteins such as GKAP[14] and SPAR[15]. It has been shown that the SH3 domain of MAGUKs has an atypical binding specificity to the GUK domain. Although this SH3 binding can occur intra/inter-molecularly, it is believed that the intramolecular mode is preferred[16,17]. Though not well understood, it is thought that the intramolecular interaction of SH3 domain is supported by other tertiary interactions when the SH3 and GUK domains are adjacent in the same polypeptide[12,18].

SAP97/hDLG is a crucial member of the DLG family of proteins involved in membrane scaffolding and activity-dependent cell morphology changes[19]. Several isoforms of SAP97 and its human orthologue hDLG (human Disc large) have been identified, making it one of the most spliced members in the family of DLGs'. These isoforms contribute to the differential expression and targeting of this protein to different subcellular regions[20,21]. SAP97/hDLG exists either as a membrane-bound α- or β-isoform characterized by the presence of Lin2-Lin7 domains. An alternatively spliced proline-rich insertion called $I_1$ is located between the N-terminal region of SAP97/hDLG and the first PDZ domain. A highly flexible intrinsically disordered "HOOK" region, separating SH3 and GUK domains has been characterized to contain two alternatively spliced insertions, namely $I_2$ and $I_3$[14,20]. In the same region, a third alternatively spliced insertion has been described as $I_4$, a brain specific isoform of SAP97/hDLG. The region separating insertion sites of $I_2/I_3$ and $I_4$ is also alternatively spliced and given the name $I_5$[20]. $I_3$ and PDZ 1-2 domains of SAP97/hDLG show similarly charged residues, forming binding sites for 4.1-like proteins[14]. These sites contribute to SAP97/hDLG localization at the sites of cell-cell contact. $I_3$ is also known to be responsible for localizing the protein to the plasma membrane[21,22]. $I_2$ is reported to be responsible for targeting SAP97/hDLG to the nucleus[20], though contradictory results have been documented[21,23]. The intrinsically disordered HOOK region of SAP97/hDLG responds to elevated intracellular $Ca^{2+}$ by interacting with the $Ca^{2+}$-binding protein Calmodulin (CaM). In the absence of intermolecular interaction, the HOOK region masks protein–protein interaction sites in the SH3 region of SAP97/hDLG[14,24]. Interaction of the HOOK region of SAP97/hDLG to CaM alters its molecular conformation, thereby destabilizing the interaction between SH3 and GUK domains[12,25–27]. In vivo, and in vitro studies imply that binding of CaM or any other high-affinity ligands to the HOOK region can destabilize SAP97 as well as other MAGUKs[1,25,27]. Destabilizing

intramolecular SH3-GUK interaction can facilitate the rapid formation of multi-protein complexes, further aiding PDZ interacting receptors to get immobilized on the membrane[4]. Among the different SAP97/hDLG isoforms, CaM has been confirmed to be associated only with the HOOK regions of $I_3$ containing isoforms[28]. It is not well understood if the different C-terminal spliced variants of SAP97/hDLG respond differentially to $Ca^{2+}$-bound CaM ([$Ca^{2+}$.CaM])[20,21].

Over the last decade, our understanding of the formation, maintenance, and elimination of molecular clusters at nanoscale is getting re-evaluated vigorously in vitro and in vivo. Growing body of evidence confirms the existence of regulatory nanodomains, which are formed transiently in real-time at spatial scales of 10–200 nm in cells[29,30]. In parallel, many of these molecules involved in the formation of transient domains also participate in liquid–liquid phase separation (LLPS)[31–33]. In vitro and in vivo evaluation of LLPS also indicate the presence of nano to micron sized phase separated compartments with the potential to have altered kinetics of exchange within and outside such compartments[34,35]. In vivo, such differing kinetics arise as a response to cellular environment, modification that happens post translationally or because of multicomponent system within a cell. To get a better insight into this regulation in the context of SAP97/hDLG, we performed direct stochastic optical reconstruction microscopy (dSTORM), a super resolution technique, to understand the endogenous distribution of SAP97/hDLG in heterologous cells. We observed that nanoscale segregation of SAP97/hDLG was physiologically regulated by the availability of free $Ca^{2+}$ and modulated through the $Ca^{2+}$ sensing protein Calmodulin. Using single-molecule localization microscopy in combination with $Ca^{2+}$ perturbation and analysis paradigms to evaluate the free energy change in protein aggregation, we show that the nanoscale condensation of SAP97/hDLG followed a first-order phase transition with spontaneous nucleation and growth. These nanocondensates displayed an altered phase transition in response to differential modulation of intracellular $Ca^{2+}$ levels. Since SAP97/hDLG can associate with [$Ca^{2+}$.CaM], we confirmed the presence of its C-terminal spliced isoforms in the hippocampal and cortical regions of rodent brains as well as in heterologous cell lines. These isoforms are known to alter the dynamics of several cell surface molecules including cell adhesion molecules, channels, and receptors such as AMPA receptors, which are known to form distinct subsynaptic nanodomains[36,37]. Next, using Fluorescence Recovery after Photobleaching (FRAP) in combination with Total Internal Reflection Fluorescence (TIRF) illumination in live neuroblastoma cells, we evaluated the exchange of SAP97/hDLG isoforms in a diffraction limited compartment. The rate of exchange of alternatively spliced C-terminal variants of SAP97/hDLG were distinct from each other. This local exchange of each isoform was influenced by cytoplasmic $Ca^{2+}$ levels and was modulated by the interaction of the intrinsically disordered HOOK region of SAP97/hDLG to $Ca^{2+}$-binding proteins. Furthermore, in vitro phase transition studies confirmed that the isoforms of SAP97/hDLG can transition into phase separated condensates and can co-condense Calmodulin in a $Ca^{2+}$-dependent manner. We confirmed that these differential effects of isoform expression at micron scale is also conserved in vivo at nanoscale using dissociated hippocampal neurons and neuroblastoma cells.

Here, with the aid of a paradigm to evaluate the molecular fingerprints of first-order phase transition across spatial scales in vivo and in vitro, we confirm that the local compositionality of isoform expression has a direct consequence on the real-time local exchange and nanoscale condensation of SAP97/hDLG. Additionally, our results verify that cellular condensation of SAP97/hDLG is differentially regulated between its isoforms, and

this variability is regulated by the spatial heterogeneity and exchange kinetics of isoforms present in near membrane SAP97/hDLG condensates.

## Results

**Nanoscale organization of SAP97/hDLG follows a first-order phase transition**. SAP97/hDLG is involved in several near membrane signaling complexes and is an integral scaffold for many transmembrane molecules. The local heterogeneity of SAP97/hDLG is critical for its association with different membrane-associated signaling complexes. To understand the fine variability of SAP97/hDLG nanoscale organization, we relied on direct stochastic optical reconstruction microscopy (dSTORM), which allows the reconstruction of a sub-diffraction limited image from several single-molecule localizations[29,38,39]. Neuro-2a cells were immunolabelled for SAP97/hDLG and

imaged by dSTORM, revealing endogenous clusters of SAP97/hDLG. Widefield conventional epifluorescence illumination showed diffused distribution of SAP97/hDLG across different cellular compartments. dSTORM revealed a large population of sub-diffraction sized molecular aggregates of SAP97/hDLG (Fig. 1A). Recent efforts in extracting morpho-functional clustering parameters from single-molecule localization such as number of molecules, length, size, and isotropic distribution of molecules in clusters have allowed to model this distribution as a first-order phase transition[34]. We followed similar paradigms to extract these individual parameters from super resolution images and model the nanoscale heterogeneity observed as phase transitions for endogenous distribution of SAP97/hDLG. First, we estimated the number of SAP97/hDLG molecules per cluster ($n$) by ratiometric analysis between ensemble fluorescence intensity of clusters and isolated single molecules outside the clusters[40,41]. The evaluation paradigms that we used for quantifying the

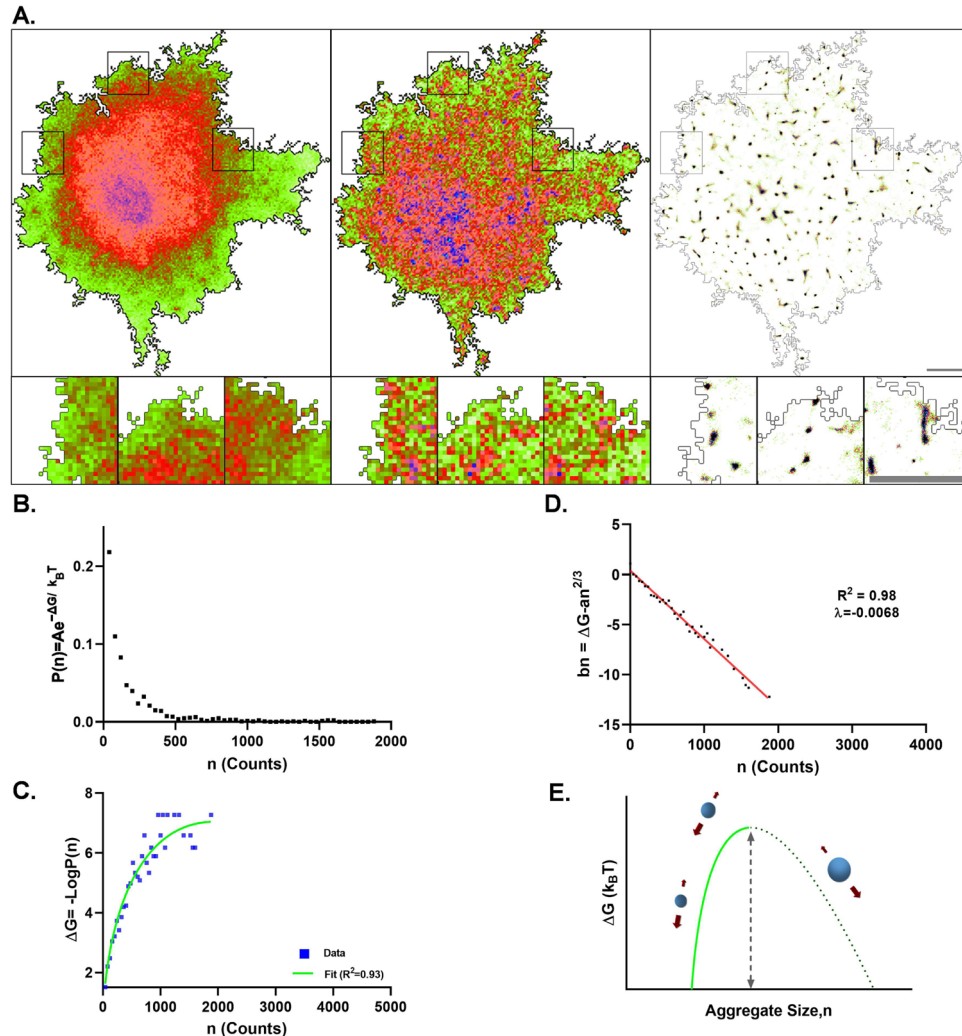

**Fig. 1 Nanoscale lateral organization of SAP97/hDLG follows a first-order phase transition in Neuro-2a cells. A** Conventional epifluorescence microscopy, Total internal reflection (TIRF) microscopy and direct stochastic optical reconstruction microscopy (dSTORM) images of SAP97/hDLG in Neuro-2a cell showing the sub-diffracted population of clusters. **B** A plot of the probability distribution function of single molecules detected inside the nanoclusters. **C** The curve fit of the inverse of probability distribution of molecules according to the function $an^{2/3} - bn + c$ to obtain the parameters $a$, $b$, and $c$, which define the nucleation barrier ($\Delta G_c$) and critical cluster radius ($R_c$). **D** Linear regression plot of the resultant surface energy correction by subtracting the surface energy ($an^{2/3}$) from the $-\mathrm{Log}P(n)$ data. The negative linear slope indicates the second term b, which corresponds to an increase in entropy. **E** Schematic of the free energy function, which follows a first-order phase transition. The broad arrows represent the favored direction of aggregate size with respect to spontaneous growth or diffusion of aggregates. The solid line represents the extent of experimental values. The critical cluster size and nucleation barrier height were calculated from the curve fit. $N = 1297$ clusters from 9 cells. Cumulative data from three experiments. All cells were fixed before imaging. Log refers to the natural logarithm to base "e". Scalebar indicates 4 μm.

molecular content resulted in extracting similar copy numbers of selected molecules as reported previously, validating the robustness of the workflow[40]. Distribution of the number of molecules detected inside a cluster resulted in a histogram where the probability of occurrence decreases rapidly towards clusters with very high molecular content behaving non-uniformly as supramolecular aggregation patterns (Fig. 1B). The free energy ($\Delta G(n_c)$) was determined from the distribution of cluster sizes, which was fit by the inverse of the probability distribution of molecules, allowing a direct estimation of the nucleation barrier ($\Delta G_c$) and critical cluster radius ($Rc$) (Fig. 1C). Saturation state of the system was defined by the values of SAP97/hDLG ambient monomer concentration ($C_{amb}$) and saturation concentration ($C_{sat}$) at equilibrium with the clustered phase. We then extracted the number of detected SAP97/hDLG molecules per nanoscale both from high-density molecular domains, which depicted zones of functional aggregation (nanodomains) and disperse molecules from extra-nanodomain regions. A positive ($C_{amb} < C_{sat}$) or negative ($C_{amb} > C_{sat}$) sign depicted the cluster's saturation state, i.e., whether the system was in a sub- or a super-saturated state. Bulk energy of the system was extracted from the data by subtracting the surface energy from free energy, and the curve was observed to be linear with a negative slope, representing a supersaturated system (Fig. 1D). $P(n)$ represents the modeled relative frequency distribution of cluster size ($n$), and $n_c$ is the critical cluster size attaining a maximum value of $\Delta G(n_c)$, which is the nucleation barrier (Fig. 1E). The data allowed us to extract the nucleation barrier $\Delta G(n_c)$ from the polynomial fit, which was normalized ($\Delta G(n_c) = 1.00 \pm 0.08$). The critical radius of cluster ($R_c$; $n_c \propto (R_c)^3$) of untreated control dataset was calculated and normalized ($R_c = 0.98 \pm 0.06$). These results confirmed that the SAP97/hDLG cluster sizes were below the diffraction limit, and the formation of these nanoscale condensates were governed by single order phase transitions, emphasizing the requirement for super-resolution microscopy techniques to resolve and measure them.

**Modulation of intracellular Ca²⁺ and Calmodulin alters the dynamics of first-order phase transition of SAP97/hDLG.** To study the effect of intracellular $Ca^{2+}$ levels on the nanoscale organization and first-order phase transition of endogenous SAP97/hDLG, we relied on a pharmacological approach to modulate cytosolic $Ca^{2+}$ levels. We incubated the cells with Thapsigargin (Tg) (1 μM)[42–44] for elevating intracellular $Ca^{2+}$ and with the membrane-permeable BAPTA-AM (Bapta) (50 μM)[45] to chelate the soluble $Ca^{2+}$ present in the cell. To confirm differential sensitivity of SAP97/hDLG to [$Ca^{2+}$.CaM], we inhibited the $Ca^{2+}$ sensing ability of CaM using w7 (25 μM)[46,47]. The aforementioned treatments altered the intracellular $Ca^{2+}$ levels and modulated the $Ca^{2+}$-mediated interaction of Calmodulin to the intrinsically disordered HOOK region of SAP97/hDLG within the cell. The consequential effect on SAP97/hDLG nucleation energetics was extracted from the analysis of single-molecule-based super resolution data as explained below.

Similar to the paradigm we followed for endogeneous distribution of SAP97/hDLG molecules, we performed dSTORM microscopy on Neuro-2a cells labeled for endogeneous SAP97/hDLG, where the cells were treated with Thapsigargin, w7 and Bapta (Fig. 2A–C). For each condition, a probability fit was performed (Fig. 2D–F) to extract the critical radius and a free energy function was plotted against the number of molecules per cluster (Fig. 2G). Interestingly, all conditions showed negative slopes for bulk energy, stating a negative entropy of the system inducing spontaneous cluster growth (Supplementary Fig. 1A–D). The slope for untreated condition (control) was found to be

−0.0068, whereas for Tg, it was −0.0033 (Supplementary Fig. 1A, B). The ambient monomer concentration ($C_{amb}$) reduced on mobilizing SAP97/hDLG by application of Tg, resulting in an overall decrease in the bulk energy. The reduction in bulk energy upon Tg incubation without modulating the critical radius confirmed a reduction of the surface energy, $an^{2/3}$. The normalized energy barrier and critical cluster radius were compared across the different conditions to that of control cells (Fig. 2H, I). We found that the elevated intracellular $Ca^{2+}$ level induced by Tg application had no significant effect on the critical radius ($R_c$ (Tg) = $1.08 \pm 0.04$ (Fig. 2I). The nucleation barrier was also unchanged between the control and Tg-treated conditions ($\Delta G$ (Tg) = $1.04 \pm 0.08$) (Fig. 2H). However, when the cells were treated with w7 to block the CaM interaction of SAP97/hDLG, the barrier height of $\Delta G$ (w7) = $1.54 \pm 0.12$ (Fig. 2H) differed from that of control cells, along with a significant increase in the critical radius, $R_c$ (w7) = $1.49 \pm 0.03$ (Fig. 2I). Application of Bapta, which chelates free $Ca^{2+}$ inside the cell, also revealed a significant change in both the nucleation barrier and critical radius ($\Delta G$ (Bapta) = $2.45 \pm 0.23$ (Fig. 2H) and $R_c$ (Bapta) = $2.38 \pm 0.04$ (Fig. 2I)).

In summary, we evaluated the perturbation kinetics of first-order phase transitions under different pharmacological treatments in Neuro-2a cells that altered the intracellular $Ca^{2+}$ levels and SAP97/hDLG interaction with CaM. Increase in intracellular $Ca^{2+}$ did not change the critical cluster radius, while there was a reduction in the slope of bulk energy, suggesting a decrease in the ambient monomer concentration. In this case, unaltered free energy of the system with an increase in entropy points to a local regulation with reduced monomer fraction and enhanced population of segregated/clustered molecules. The perturbation with the CaM inhibitor resulted in a significant increase in the critical radius of the cluster as well as the nucleation barrier. In this case, the slope of bulk energy was further reduced due to an increase in saturation concentration of monomer. Furthermore, when a chelator blocked the intracellular $Ca^{2+}$, both the nucleation barrier and the critical radius increased, while the bulk energy reduced substantially. The results proposed an alteration in the clustering of SAP97/hDLG, forming numerous sub-saturated clusters. In parallel, we observed an increase in the dispersed phase by lowering the bulk energy. The model proposes a higher nucleation barrier and critical cluster radius in these conditions, implying that the cells are in a sub-saturated state, where clusters exist though they are spontaneously dispersed to form monomers.

**Nanoscale condensation of endogenous SAP97/hDLG is fine tuned by Ca²⁺-dependent first-order phase transition.** Though the phase transitions point towards an altered free energy profile, we identified the nanoscale fingerprints of these transitions using different stimulation paradigms. For this, we analyzed the variability of the different morpho-functional characteristics such as the average intensity and area of nanoclusters of SAP97/hDLG. The incubation of Neuro-2a cells with Tg (1 μM) (10 min), w7 (25 μM) (30 min) and Bapta (50 μM) (20 min) resulted in an enhancement of both average intensity (Fig. 3A–C) and area (Supplementary Fig. 2A–C) of SAP97/hDLG clusters. For peforming the rank-order analysis, all datasets from each treated condition were sampled to 80% of the corresponding set, and was sorted in ascending order. The treated sets were plotted against the untreated control set. The slope of the regression curve was chosen as an arbitrary scaling factor of a parameter corresponding to the shift in the treated data from control. The rank-order analysis of average intensity of SAP97/hDLG nanoclusters in the presence of Tg, w7 and Bapta vs. control showed higher

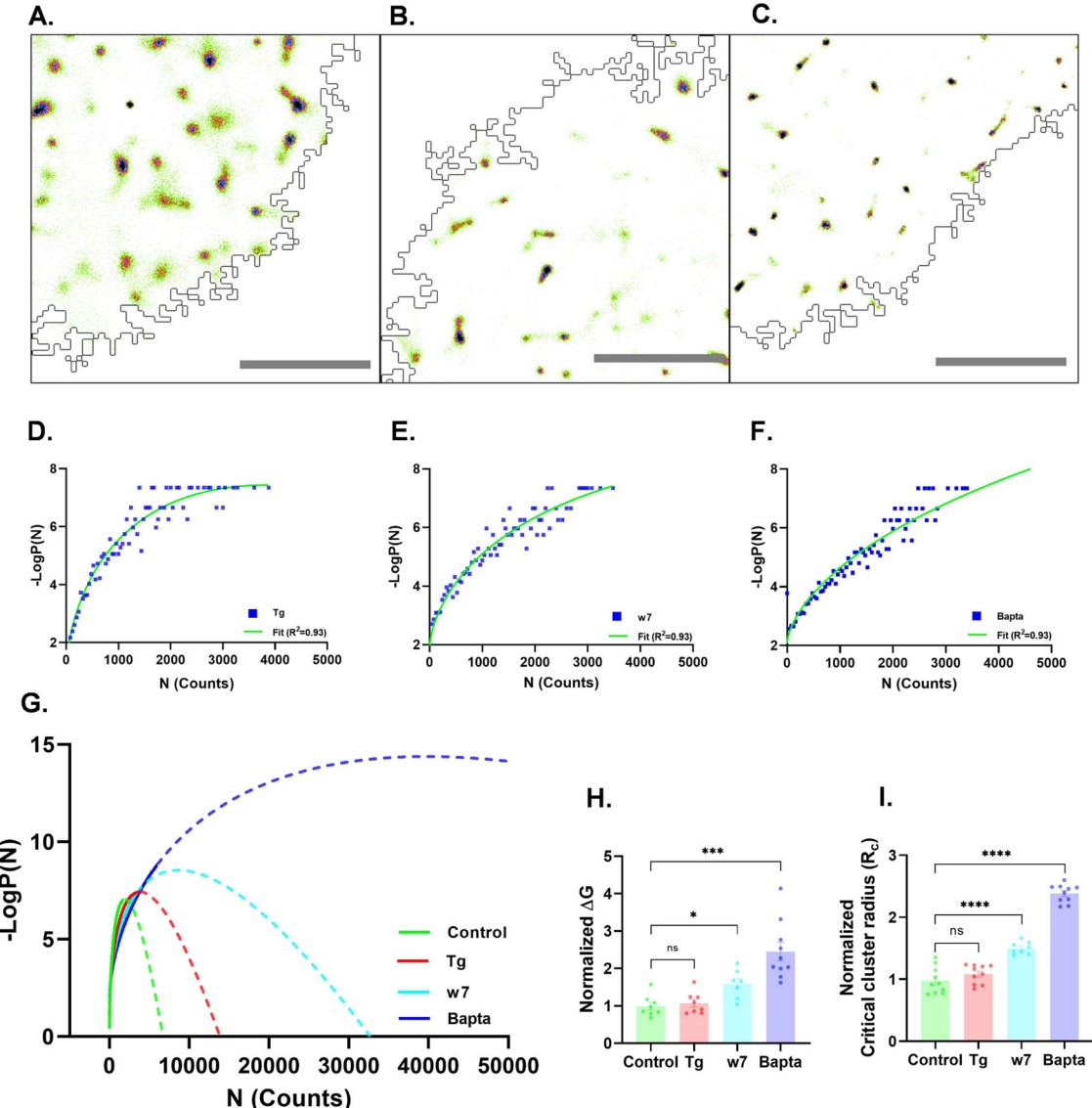

**Fig. 2 Intracellular Ca$^{2+}$ levels can regulate the super-saturation of SAP97/hDLG nanoclusters.** Reconstructed super resolved images (500 × 500 pixels) of SAP97/hDLG in Neuro-2a cells treated with **A** Thapsigargin (Tg), **B** w7, **C** Bapta, and **D–F** their –Log((n)) vs. n curve, from each condition. **G** The functional fit of –Log(P(n)) from different pharmacological treatments. **H** The normalized free energy barrier for each condition, calculated from the functional fit, is plotted and compared with control. **I** The plot represents the normalized critical cluster radius (nm) during different treatments, compared to control data. The data is from 1000–1500 clusters ($N = 10$ cells (Control, Tg and Bapta), 8 cells (w7)) in each condition cumulated from three experiments. Statistical test for $\Delta G_c$ and $R_c$ was performed by one-way ANOVA. Log refers to the natural logarithm to base "e". Data are presented as mean values ± SEM. *$p < 0.05$, ****$p < 0.0001$ (Supplementary Table 1). Scalebar = 4 µm. Source data are provided as Source Data file.

slopes of 1.67, 2.72, and 1.72, respectively (Fig. 3D–F). Then, we verified whether these data indicated an up or down scaling of the morpho-functional properties of clusters[48] in these different conditions. The actual scaling factor of the average intensity of nanodomains in the treated vs. control cells were estimated. For the purpose, the datapoints from each treated condition (Tg/w7/Bapta) was divided by 100 values within the range of the arbitrary scaling factor, keeping the arbitrary value as the median. The resultant datasets were named as scaled-treated data. To determine the actual scaling factor, the control and the scaled-treated datasets were tested for significance using Kolmogorov–Smirnov test (K–S test)[48]. The actual scaling factor should give highest $p$-value, as the difference in the datasets would be non-significant. The method was repeated 100 times with each dataset. The results provided us with 100 $p$-values for each condition, and they were plotted against their corresponding scaling factors (Fig. 3G–I).

The scaling factor corresponding to the highest $p$-value was chosen as the actual scaling factor. The resultant values of actual scaling factors for Tg, w7, and Bapta incubations were 1.66, 3.10, and 3.76, respectively (Fig. 3G–I). Thus, the scaling factors showed an upscaling of the average intensity of SAP97 in these clusters. The cumulative frequency distribution (CFD) of the average intensity of control, treated, and scaled-treated datasets were plotted against the average intensity (Fig. 3J–L). When the treated datasets were downscaled by the actual scaling factor, the scaled-treated datasets replicated the control data. This scaling was confirmed by the plot of the CFD curve of average intensity (Fig. 3J–L). In parallel, we studied the effect of Ca$^{2+}$ perturbation on the area of SAP97/hDLG clusters. The frequency distribution histograms were plotted for the treated and control conditions (Supplementary Fig. 2A–C). A total of 1000 cluster area data-points were chosen randomly from each dataset and rank-

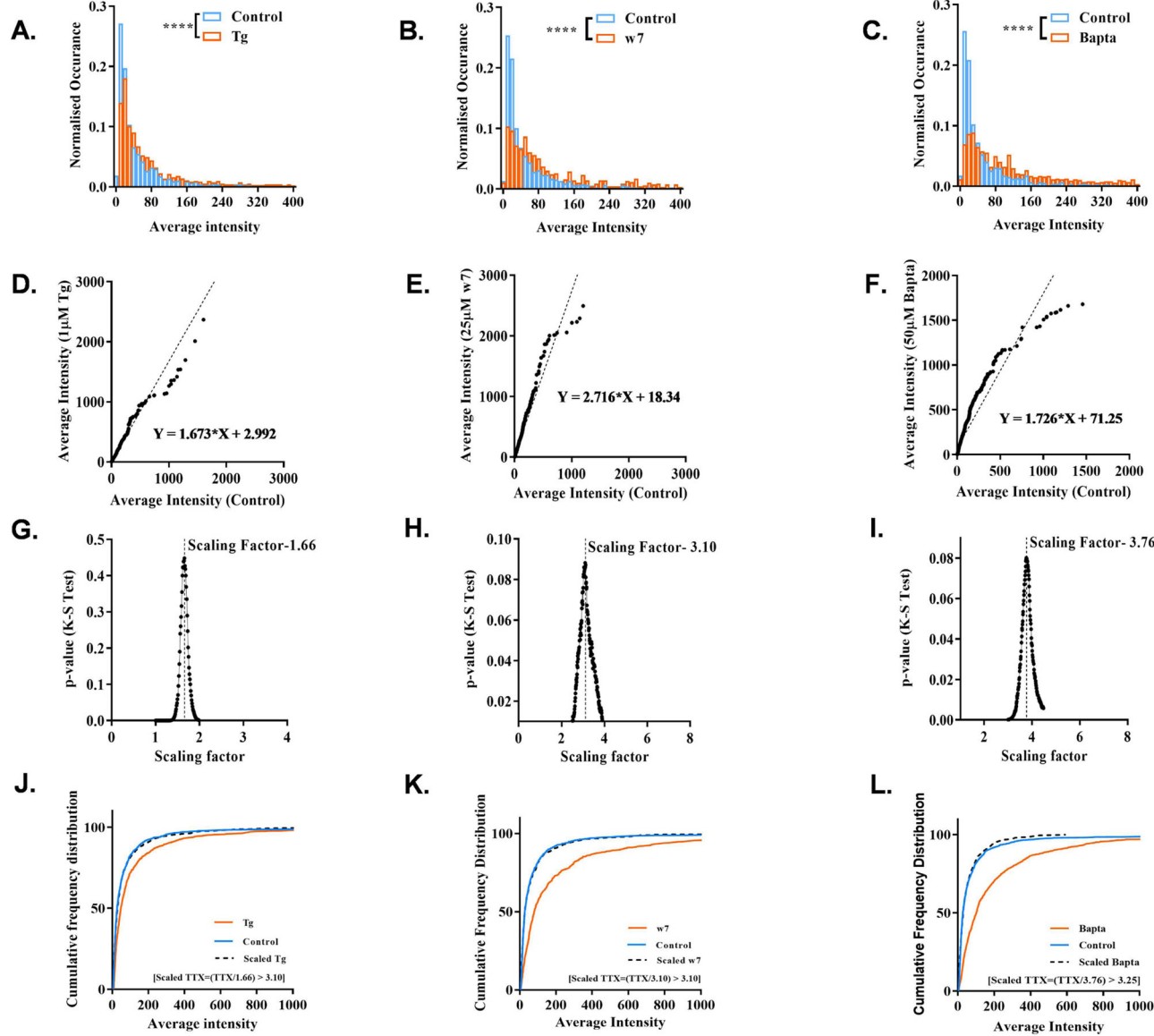

**Fig. 3 Scaling of endogenous SAP97 clusters using rank-order analysis. A–C** The histogram was plotted for the average intensity of SAP97/hDLG clusters between control and all treated conditions. **A** Thapsigargin (1 μM), **B** w7 (25 μM), and **C** Bapta (50 μM). $N = 10$ cells for control and treated datasets. **D–F** A total of 1000 cluster datapoints of average intensity were chosen randomly from each dataset, rank-ordered, and plotted to provide a linear regression for each condition. The slope of regression depicts an arbitrary scaling factor. **G–I** The analysis provided a scaling factor with a maximum $p$-value between control and scaled-treated datasets. The $p$-values for Tg (**D**), w7 (**E**), and Bapta (**F**) were $p = 0.44$, $p = 0.09$, and $p = 0.08$, respectively. A total of 800–1000 random average intensity datapoints were chosen from each dataset, rank-ordered, and plotted to provide a linear equation for each treatment. **J–L** A cumulative frequency distribution was plotted for control, treated, and scaled-treated conditions of SAP97/hDLG in Neuro-2a cells. The datasets were analyzed for a significant change in mean values by two-tailed K–S test, *$p < 0.05$, **$p < 0.01$, ****$p < 0.0001$ (Supplementary Table 2).

ordered. The rank-ordered data of cluster area was plotted for treated vs. control to determine the linear regression for the different conditions (Supplementary Fig. 2D–F). The scaling factors were derived, following the analysis for Tg (1.24), w7 (1.22) and Bapta (1.46) (Supplementary Fig. 2G–I). The CFD of cluster area of SAP97/hDLG was plotted with control, treated, and scaled-treated conditions. The CFD of scaled-treated data identically overlapped with that of the control data, verifying a significant accuracy of the calculated scaling factor.

To summarize, thapsigargin application resulted in an upscaling of average intensity and area of clusters by factors, 1.66 and 1.24, respectively. However, the nucleation theory analysis revealed no significant shift in the free energy or critical radius of SAP97/hDLG clusters in the presence of high-intracellular

$Ca^{2+}$ levels (Fig. 2H–I). Together, the results confirmed that an increase in intracellular $Ca^{2+}$ scaled up the cluster area as well as the number of SAP97/hDLG molecules inside the clusters, keeping the nucleation barrier stable. The surface energy is controlled by the number of molecules contained in the cluster and their density within the cluster, while the bulk energy is contributed by the monomer concentration outside the cluster. The observed increase in the area (Supplementary Fig. 2A) and average intensity of clusters (Fig. 3A) without affecting the free energy change but resulting in a decrease in the bulk energy (Supplementary Fig. 1B) suggested an equivalent decrease in surface energy. Here, more than the increase in molecules inside clusters, the intracluster molecular organization determines the surface energy. On application of CaM blocker w7, endogenous

SAP97/hDLG aggregated as clusters potentially via a CaM independent mechanism. This occured with a slight yet significant increment in the area of clusters by a factor of 1.22 (Supplementary Fig. 2H), but with substantial scaling of the average intensity of clusters by a factor of 3.10 (Fig. 3H). The inhibition of basal CaM activity led to a significant increase in the number of molecules inside the clusters as well as the monomer concentration outside clusters, resulting in an increase in the nucleation barrier and critical radius (Fig. 2H, I). w7 incubation resulted in a reduction in the bulk energy (Supplementary Fig. 1C). An increment in $C_{amb}$ or a decrement in $C_{sat}$ would reduce the difference in free energy between the ambient state of monomer fraction and the saturated state of monomer concentration, organizing higher number of molecules within the clusters with minute scaling in cluster area.

The cells deprived of intracellular $Ca^{2+}$ in the presence of Bapta displayed larger clusters scaled to 1.46, with the average intensity scaled to 3.76. The results suggested a strong upscaling of the number of molecules within SAP97/hDLG clusters, with a limited scaling of their cluster size on chelating intracellular $Ca^{2+}$. A substantial increase in the nucleation barrier and the critical radius (Fig. 2H, I) with reduced slope of bulk energy (Supplementary Fig. 1D) concur with a significant increase in the average intensity with slight increment in the cluster area, in the absence of $Ca^{2+}$. Together, by blocking the $Ca^{2+}$ sensor interacting with SAP97/hDLG, we observed an increase in the number of molecules included in the cluster with a slight increase in the cluster area. Alternatively, the number of SAP97/hDLG molecules as well as the area and critical radius of the clusters were scaled up enormously with reduced intracellular $Ca^{2+}$. This concurs to the existence of a $Ca^{2+}$ independent SAP97/hDLG clustering mechanism, with potential isoform-specific interactions within the endogenous pool of Neuro-2a cells. Therefore, we were interested in studying the effect of altered intracellular $Ca^{2+}$ levels on the aggregation of endogenous SAP97/hDLG and its isoforms at nanoscale level in Neuro-2a cells. Our observations affirmed that modulation of $Ca^{2+}$ or Calmodulin may not be critical for nucleation of nanoscale condensates, but necessary for their regulation.

**Variability of intrinsically disordered region of SAP97/hDLG isoforms results in differential spatio-temporal fingerprints.** Among the different isoforms expressed in neuronal cells, α- and β-isoforms of SAP97/hDLG have been well characterized[37,49]. To confirm the presence of other known isoforms of SAP97/hDLG, we performed real-time quantitative PCR experiments in the whole cell RNA extracts from cortical and hippocampal neurons of new born mice pups (P0). The fold-change in expression of each SAP97/hDLG spliced variant was normalized to the expression of β-2-microglobulin ($B_2M$) and double normalized to the cortical expression. The N-terminal and C-terminal spliced variants of SAP97/hDLG are $I_{1A}$, $I_{1B}$, and $I_2$, $I_3$, respectively (Supplementary Fig. 3A). The relative mRNA expression of SAP97/hDLG in cortex, hippocampus and Neuro-2a cells showed high variability. All SAP97/hDLG variants except $I_2$, showed low mRNA levels in Neuro-2a cells, relative to cortex and hippocampus (Supplementary Fig. 3B–G) ($N = 3$). $I_{1AB}$ isoform was found to include both $I_{1A}$ and $I_{1B}$ isoforms of the N-terminal $I_1$ splicing. $I_2$ and $I_3$ splicing were observed to be present in comparable levels in both hippocampus and cortex. We did not find a case were both $I_2$ and $I_3$ splicing coexisted, consistent with previous observations stating that a construct with both $I_2$ and $I_3$ resulted in a truncated product by induction of a stop codon after $I_3$[20]. It has been previously reported that all isoforms of SAP97/hDLG are known to express an additional C-terminal splicing in

the HOOK region, referred to as $I_5$ splicing, except for the prematurely terminated $I_4$ isoform, which is expressed in very low amounts only in the mammalian brain[20,50]. Additionally, the non-membrane-bound β-isoform (Supplementary Fig. 3A) accounts for about 90% of ubiquitously expressed SAP97/hDLG. However, antibodies specific for SAP97/hDLG variants are not available because of the difficulty in developing isoform-specific antibodies. Thus, we relied on the ectopic expression of SAP97/hDLG isoforms containing a combination of β-$I_{1AB}$-$I_2$-$I_5$ splicing (referred to as $I_{1AB}$-$I_2$-$I_5$) or β-$I_{1AB}$-$I_3$-$I_5$ (referred to as $I_{1AB}$-$I_3$-$I_5$) to quantify the protein trafficking dynamics in live cells.

The exchange kinetics of SAP97/hDLG isoforms were evaluated by assessing the ensemble recovery of molecules in a diffraction limited region corresponding to the point spread function. Fluoresence recovery after photobleaching (FRAP) experiments in total internal reflection fluorescence (TIRF) mode were conducted on Neuro-2a cells after ectopically expressing GFP:: $I_{1AB}$-$I_2$-$I_5$ or GFP:: $I_{1AB}$-$I_3$-$I_5$ constructs (Supplementary Fig. 4A). The fluorescence recovery inside a region of interest (ROI) (Supplementary Fig. 4B) was measured to calculate the mobile fraction as well as the halftime ($t_{1/2}$) of hDLG isoforms. Confocal microscopy revealed an isoform-specific difference in the localization of the protein in cells (Supplementary Fig. 5A, B). The mobile fraction and halftime of recovery from FRAP analysis revealed a significant variation between the isoforms (mobile fraction; $I_2 = 0.74 \pm 0.02$, $I_3 = 0.62 \pm 0.04$ (Supplementary Fig. 6B, C) and halftime of recovery; $I_2 = 0.25 \pm 0.03$ s, $I_3 = 0.22 \pm 0.03$ s (Supplementary Fig. 6D)). These observations verified that the behavior of $I_3$ isoform is distinct from that of $I_2$ at the near membrane region of Neuro-2a cells. This observation of altered mobile fraction of isoforms motivated us to investigate the trafficking dynamics of both isoforms by modulating the intracellular $Ca^{2+}$, thereby mimicking activity-dependent physiological changes.

**Differential binding of $I_2$ and $I_3$ SAP97/hDLG variants to $Ca^{2+}$-bound CaM modulate their exchange kinetics.** The fluorescence recovery curves of both $I_2$ & $I_3$ isoforms displayed significant differences in protein mobility when $Ca^{2+}$ levels were perturbed. The application of Tg did not affect the mobile fraction of $I_2$ ($I_2 - Tg = 0.75 \pm 0.02$, $I_2 = 0.74 \pm 0.02$), while the halftime of recovery was increased ($I_2 - Tg = 0.35 \pm 0.03$ s, $I_2 = 0.25 \pm 0.03$ s) (Fig. 4A). CaM being a known binding partner of SAP97/hDLG, cells expressing $I_2$ were treated with CaM blocker w7, which revealed a significant reduction in its mobile fraction ($I_2 - w7 = 0.60 \pm 0.02$, $I_2 = 0.74 \pm 0.02$) (Fig. 4B) along with a significantly reduced halftime of recovery ($I_2 - w7 = 0.09 \pm 0.01$ s, $I_2 = 0.25 \pm 0.03$ s) (Fig. 4C). Decrease in the mobile fraction of $I_2$ molecules in the absence of CaM activity points to a CaM-dependent control on its exchange kinetics in resting cells. Incubating cells with Bapta to reduce free intracellular $Ca^{2+}$ resulted in reducing the mobile fraction of the $I_2$ isoform ($I_2 - Bapta = 0.66 \pm 0.02$, $I_2 = 0.74 \pm 0.02$) (Fig. 4B). In this case, the halftime of the molecules showed a significant increase, revealing an extensive trapping of $I_2$ molecules near the membrane ($I_2 - Bapta = 1.03 \pm 0.04$ s, $I_2 = 0.25 \pm 0.03$ s). The reduced recovery and enhanced halftime demonstrated a high-bound fraction of $I_2$-hDLG upon reducing cytosolic $Ca^{2+}$ levels in cells.

In the case of $I_3$, Tg application resulted in a higher mobile fraction ($I_3 - Tg = 0.66 \pm 0.04$, $I_3 = 0.50 \pm 0.02$) with a shortened halftime of recovery ($I_3 - Tg = 0.08 \pm 0.01$ s, $I_3 = 0.21 \pm 0.02$ s) (Fig. 4D). The results suggested that the mobility of $I_3$ isoform is dependent on the intracellular $Ca^{2+}$ concentration. CaM inhibition using w7 increased the mobile fraction ($I_3 - w7 = 0.56 \pm 0.02$, $I_3 = 0.50 \pm 0.02$) (Fig. 4E), but not the halftime of

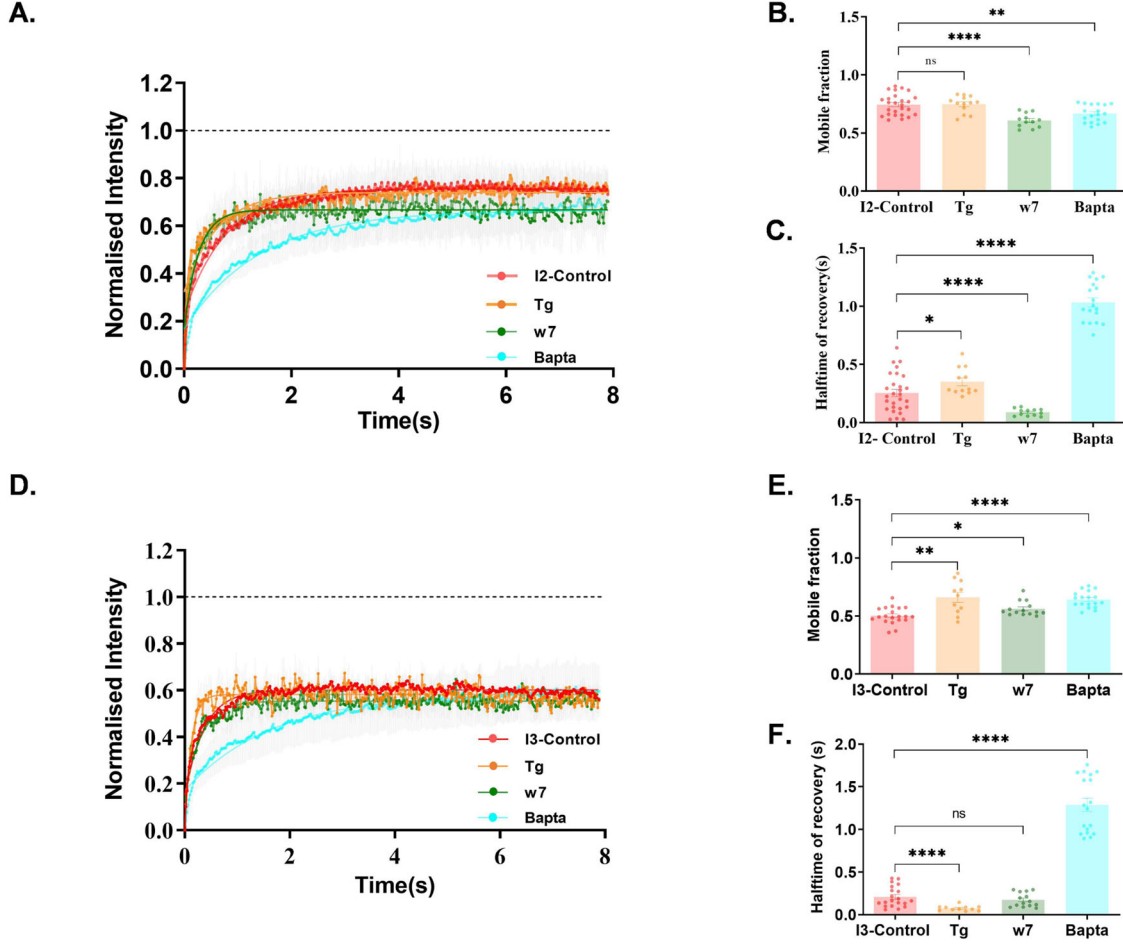

**Fig. 4 Differential dynamics of hDLG::GFP variants upon modulating cytosolic Ca²⁺ in Neuro-2a cells.** Cells expressing $I_2$-hDLG::GFP (**A**–**C**) and $I_3$-hDLG::GFP (**D**–**F**) were treated with Tg, w7, and Bapta (**A**, **D**). The recovery of fluorescence intensity against time is plotted for different conditions. Curve fitting was done using a single-phase exponential growth curve. **B**, **C**, **E**, **F** The mobile fraction and halftime of recovery were recovered from the respective curves and tested by one-way ANOVA. $N = 13$ cells; 26 ROI (Control), 8 cells; 13 ROI (Tg), 8 cells; 12 ROI (w7), and 9 cells; 18 ROI. The data was cumulated from three experiments. Data are presented as mean values ± SEM. *$p < 0.05$, **$p < 0.01$, ****$p < 0.0001$ (Supplementary Table 3). Source data are provided as Source Data file.

recovery ($I_3 - w7 = 0.17 \pm 0.02$ s, $I_3 = 0.21 \pm 0.02$ s) (Fig. 4F). The results indicated a CaM dependant modulation of the mobile fraction of the $I_3$ isoform. However, the cells incubated with Bapta displayed a significant increase in the mobility of $I_3$ ($I_3 - Bapta = 0.64 \pm 0.01$, $I_3 = 0.50 \pm 0.02$) with an extended half-time of recovery ($I_3$-Bapta = $1.28 \pm 0.07$ s, $I_3 = 0.20 \pm 0.02$ s), similar to the $I_2$ isoform (Fig. 4E).

To confirm the modulation of the $I_2$ isoform by Ca²⁺-bound Calmodulin, the cells were treated with w7 followed by Tg. The results showed a compelling difference in the mobile fraction of $I_2 - Tg + w7$ ($I_2 - Tg + w7 = 0.60 \pm 0.01$, $I_2 - Tg = 0.74 \pm 0.02$), while the halftime was significantly increased ($I_2 - Tg + w7 = 0.55 \pm 0.04$, $I_2 - Tg = 0.35 \pm 0.03$ s) (Fig. 5A), substantiating a CaM independent or an intrinsic clustering mechanism for the $I_2$ isoform. To understand the role of intracellular Ca²⁺ stores and store operated Ca²⁺ entry (SOCE) on the dynamics of SAP97/hDLG isoforms, the cells were treated with Tg followed by Bapta. The cells expressing $I_2$ showed no major shift in their mobile fraction ($I_2 - Tg + BAPTA = 0.76 \pm 0.01$, $I_2 - Tg = 0.74 \pm 0.02$) (Fig. 5B), while there was a significant difference in their halftime, revealing a slower recovery ($I_2 - Tg + Bapta = 0.47 \pm 0.04$ s, $I_2 - Tg = 0.35 \pm 0.03$ s) (Fig. 5C). Cells expressing $I_3$ in the presence of both w7 and Tg displayed no significant change in their mobile fraction ($I_3 - Tg + w7 = 0.55 \pm 0.02$, $I_3 - Tg = 0.63 \pm 0.05$) (Fig. 5E),

while the halftime of recovery was delayed significantly ($I_3 - Tg + w7 = 0.38 \pm 0.02$, $I_3 - Tg = 0.07 \pm 0.01$) (Fig. 5F). When the cells were deprived of free Ca²⁺ after inducing SOCE, the $I_3$ isoform showed a slower recovery ($I_3 - Tg + Bapta = 0.85 \pm 0.04$ s, $I_3 - Tg = 0.07 \pm 0.01$ s) without a significant difference in its mobile fraction ($I_3 - Tg + Bapta = 0.55 \pm 0.02$, $I_3 - Tg = 0.63 \pm 0.05$), compared to Tg application (Fig. 5D–F). These results confirmed that SAP97/hDLG isoforms respond to alteration in cytosolic Ca²⁺ and modulation of Calmodulin differentially. Additionally, these experiments show augumented levels of $I_2$ and $I_3$ isoforms in our experimental system, and indicate that their variability in exchange kinetics might affect the nucleation barrier and critical cluster radius, opening up a role for compositionality of these isoforms in controlling local signatures of SAP97/hDLG phase transition.

On comparing the distinct recovery dynamics of SAP97/hDLG isoforms, the mobile fraction of $I_2$ was significantly higher with respect to $I_3$ ($I_2 = 0.74 \pm 0.01$; $I_3 = 0.50 \pm 0.01$). $I_2$ applied with Tg, w7, and Bapta showed identical levels of mobile fraction compared to $I_3$ ($I_2 - Tg = 0.74 \pm 0.02$; $I_3 - Tg = 0.79 \pm 0.03$, $I_2 - w7 = 0.60 \pm 0.01$; $I_3 - w7 = 0.56 \pm 0.01$, $I_2 - Bapta = 0.66 \pm 0.01$; $I_3 - Bapta = 0.64 \pm 0.01$) (Fig. 5G). However, Tg treatment following w7 application resulted in an isoform-specific distinction in the response of their mobile fraction ($I_2 - Tg + w7 = 0.60 \pm 0.01$; $I_3 - Tg + w7 = 0.55 \pm 0.02$). When SOCE was induced followed

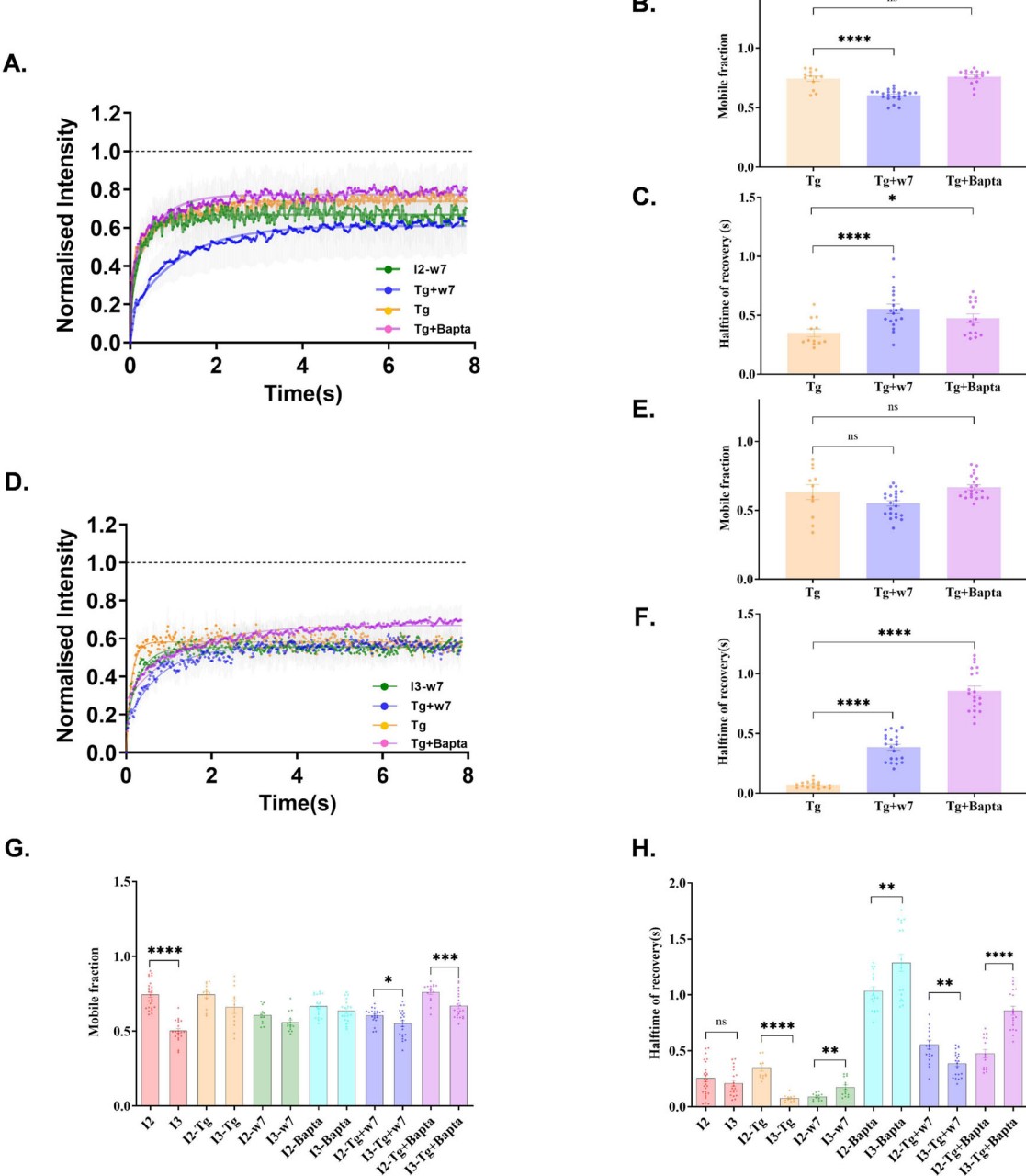

**Fig. 5 Effect of Calmodulin and cytosolic Ca$^{2+}$ levels on hDLG::GFP variants.** The recovery of fluorescence intensity in cells expressing $I_2$-hDLG::GFP (**A**) and $I_3$-hDLG::GFP (**D**) treated with w7, Tg, w7 + Tg, and Tg + Bapta fit by a single-phase exponential growth curve. $N = 8$ cells; ROI = 13 (Tg), 10 cells; ROI = 22 (Tg + w7), 9 cells; ROI = 15. The data is cumulated from three experiments. The data was statistically tested using a two-tailed unpaired $t$-test with Welch's correction. *$p < 0.05$, **$p < 0.01$, ****$p < 0.0001$ (Supplementary Table 4). The mobile fraction and halftime of recovery of $I_2$ (**B**, **C**) and $I_3$ (**E**, **F**) were extracted from the analysis of the respective curves. Mobile fraction (**G**) and halftime of recovery (**H**) of all conditions between $I_2$-hDLG::GFP and $I_3$-hDLG::GFP expressing cells were analyzed using a two-tailed unpaired $t$-test with Welch's correction. Data are presented as mean values ± SEM. *$p < 0.05$, **$p < 0.01$, ****$p < 0.0001$ (Supplementary Tables 4 and 5). Source data are provided as Source Data file.

by Ca$^{2+}$ inhibition, $I_3$ was found to be significantly confined with respect to $I_2$ ($I_2 − \text{Tg} + \text{Bapta} = 0.76 \pm 0.01$; $I_3 − \text{Tg} + \text{Bapta} = 0.67 \pm 0.02$) (Fig. 5G). The halftime of recovery showed no significant isoform-specific difference in untreated condition ($I_2 = 0.25 \pm 0.03$; $I_3 = 0.21 \pm 0.03$), while with Tg incubation, $I_3$ showed faster kinetics with respect to $I_2$ ($I_2 − \text{Tg} = 0.35 \pm 0.03$; $I_3 − \text{Tg} = 0.07 \pm 0.01$). CaM blockade with w7 shifted $I_2$ towards faster kinetics compared to $I_3$ ($I_2 − \text{w7} = 0.08 \pm 0.01$ s; $I_3 − \text{w7} = 0.17 \pm 0.02$ s). In the presence of Bapta, though there was a significant delay in the halftime for both isoforms, $I_2$ showed faster kinetics than $I_3$

($I_2 – \text{Bapta} = 1.03 \pm 0.04$ s; $I_3 – \text{Bapta} = 1.30 \pm 0.07$ s) (Fig. 5H). Elevation of intracellular Ca$^{2+}$ following CaM inhibition resulted in $I_3$ displaying faster kinetics compared to $I_2$ ($I_2 − \text{w7} + \text{Tg} = 0.55 \pm 0.04$ s; $I_3 − \text{w7} + \text{Tg} = 0.39 \pm 0.02$). On the contrary, $I_2$ isoform showed a faster recovery on depleting Ca$^{2+}$ stores as well as on chelating the intracellular free Ca$^{2+}$, compared to $I_3$ ($I_2 − \text{Tg} + \text{Bapta} = 0.47 \pm 0.04$; $I_3 − \text{Tg} + \text{Bapta} = 0.85 \pm 0.04$ s) (Fig. 5H). Additionally, the different experimental conditions yielded a similar outcome independent of duration of treatment, indicating that the observed effects were saturated (Supplementary Fig. 7A–C). These

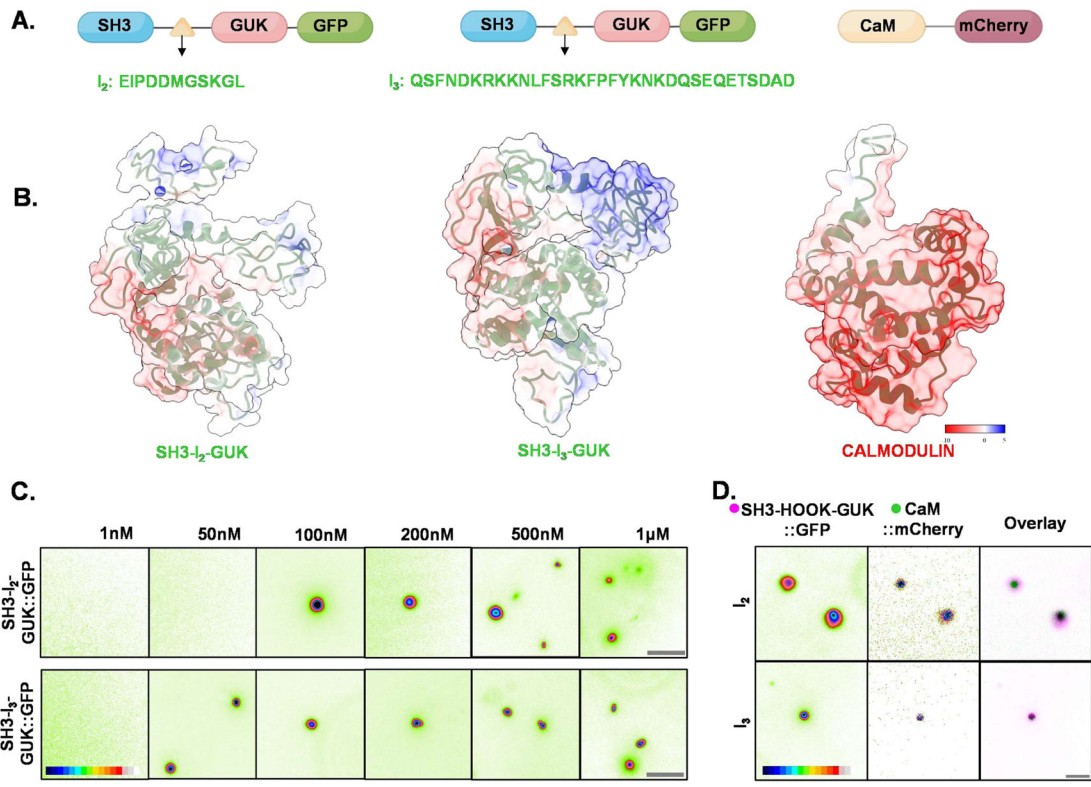

**Fig. 6 In vitro liquid–liquid phase separation of SAP97/hDLG. A** A scheme indicating purified SH3-HOOK-GUK isoforms, namely $I_2$ and $I_3$, of C-terminal fragments of SAP97/hDLG and Calmodulin flanked by enhanced green fluorescent protein and monomeric mCherry, respectively, at the C-terminus. **B** The electrostatic distribution of net surface charges of SH3-$I_2$-GUK, SH3-$I_3$-GUK and Calmodulin. Pseudo color scalebar indicates transition from positive to negative (red to blue). **C** Top and bottom panels indicate the phase separation of C-terminal isoforms, namely SH3-$I_2$-GUK::GFP and SH3-$I_3$-GUK::GFP in response to their increasing concentration at 20% PEG. SH3-$I_2$-GUK::GFP nucleation occurs at a concentration higher than SH3-$I_3$-GUK::GFP, indicating that spontaneous phase transition of $I_3$ isoform of SAP97/hDLG occurs at a lower concentration than that of the $I_2$ isoform. Scalebar indicates 5 μm. **D** Incubation of 2 μM of CaM::mCherry with 1 μM SH3-$I_2$-GUK::GFP or SH3-$I_3$-GUK::GFP in presence of 2 mM CaCl₂ resulted in the formation of co-condensates of SAP97/hDLG and Calmodulin. In the overlap images, SAP97/hDLG is in magenta and Calmodulin in green and their overlay is indicated by black. Scalebar indicates 3 μm.

results confirmed that both $I_2$ and $I_3$ isoforms are differentially influenced by Calmodulin. However, the profound effects of [Ca²⁺.CaM] was observed for the $I_3$ isoform.

**Isoform-dependent phase transitions of SAP97/hDLG and co-condensation with [Ca²⁺.CaM].** The cell is a multicomponent system where several factors affect the assembly, disassembly, and conformational state of molecules. Our results from super-resolution microscopy and FRAP indicate that SAP97/hDLG isoforms have differing kinetics of association with Calmodulin. Considering that the difference between the SAP97/hDLG isoforms is only the presence of either the $I_2$ or $I_3$ insert, we compared the electrostatic distribution of the SH3-HOOK-GUK region of SAP97/hDLG and Calmodulin (Fig. 6A). We found that the HOOK region of C-terminal fragment of SAP97/hDLG showed a net-positive charge for both $I_2$ and $I_3$, while Calmodulin displayed a negative charge in its Ca²⁺ unbound form (Fig. 6B). However, a closer evaluation of the HOOK region indicated that these charges were spread out in the $I_2$ isoform, while specifically localized in $I_3$ (Fig. 6B). This confined localization of charges in the $I_3$ isoform poses a weak attraction towards CaM. Thus, $I_3$ isoform would be a better candidate for compartmentalizing Calmodulin.

We evaluated if the alterations of nanoscale condensates and heterogeneity in the response of SAP97/hDLG isoforms to Ca²⁺ and Calmodulin was a result of their intrinsic property resulting from LLPS. We purified the C-terminal fragment of SAP97/

hDLG, which contain the SH3-HOOK-GUK region fused to GFP (Supplementary Figs. 8A–F and 9A–D), since the alternate splicing occurs in the intrinsically disordered "HOOK" region. The purified HOOK region either contained $I_2$ or $I_3$ splicing and is referred to as SH3-$I_2$-GUK::GFP and SH3-$I_3$-GUK::GFP, respectively (Fig. 6A and Supplementary Fig. 9A–D), To evaluate the homotypic phase transition of these isoforms, we increased stepwise the concentration of the crowding agent, polyethylene glycol (PEG), to 1 μM concentration of SAP97/hDLG in solution. As the concentration of PEG reached 20%, we observed a spontaneous phase separation of both SH3-$I_2$-GUK::GFP and SH3-$I_3$-GUK::GFP into micron sized condensates (Supplementary Fig. 10). Next, to examine the dependence of this separation to the concentration of the C-terminal fragment of SAP97/hDLG, we progressively increased the concentration of SAP97/hDLG, while keeping the concentration of PEG constant (Fig. 6C). While LLPS of SH3-$I_2$-GUK::GFP started to occur at 100 nM concentration, SH3-$I_3$-GUK::GFP partitioned into microscale condensates at low concentrations as 50 nM (Fig. 6C). These results confirmed the ability of SAP97/hDLG for homotypic phase transition with distinct concentration threshold for the evolution of condensates.

Next, we evaluated if Calmodulin could form independent condensates similar to SAP97/hDLG or transition into the condensates formed by LLPS. We assessed whether purified Calmodulin tagged with a monomeric variant of red fluorescent protein Cherry, referred to as CaM::mCherry, can phase separate

alone or at increasing $Ca^{2+}$ concentrations up to 2 μM. CaM::mCherry remained soluble and did not transition into micron scale condensates similar to SAP97/hDLG in any of these conditions (Supplementary Fig. 11). This observation remained consistent upon increasing its concentration at 20% PEG or on increasing the concentration of crowding agent up to 40% or on elevating $CaCl_2$ in solution to 2 mM (Supplementary Fig. 11). 1 μM SH3-$I_2$-GUK::GFP or SH3-$I_3$-GUK::GFP and 2 mM $CaCl_2$ were then co-incubated with increasing concentrations of CaM::mCherry from 100 nm to 2 μM. CaM::mCherry was recruited into SAP97/hDLG condensates when it was 2 μM for SH3-$I_2$-GUK::GFP and 1 μM for SH3-$I_3$-GUK::GFP, respectively (Fig. 6D and Supplementary Fig. 12A, B). However, in the presence of 1 μM SH3-$I_2$-GUK::GFP or SH3-$I_3$-GUK::GFP, though we could observe LLPS for SAP97/hDLG, similar concentration or increasing concentrations up to 2 μM did not yield any condensation for CaM::mCherry with SAP97/hDLG in the absence of $Ca^{2+}$ (Supplementary Fig. 13A, B). This showed that CaM requires both SAP97/hDLG and $Ca^{2+}$ for its recruitment into LLPS domains of SAP97/hDLG. Additionally, co-condensation of CaM with $I_3$ was observed at a lower concentration of CaM, as compared to $I_2$ isoform (Supplementary Fig. 12). These experiments confirmed that the SH3-HOOK-GUK region can recruit CaM into its LLPS condensates in a $Ca^{2+}$-dependent manner, which is a direct consequence arising as a result of alternative splicing of SAP97/hDLG C-terminal region and their differences in sensitivity to [$Ca^{2+}$.CaM].

One of the interesting outcomes of these experiments is that the expression of local compositionality of isoforms inside a cell could have a direct consequence on the nucleation of nanoscale condensation of SAP97/hDLG. To the best of our knowledge, there are no existing methods that allow us to label these isoforms differently in the endogenous population. Considering phase transition of SAP97/hDLG to be dependent on the properties of isoforms, neurons and Neuro-2a cells express different combinations of its C-terminal spliced variants (Supplementary Fig. 3). Consistent with previous reports[37,51], by ectopic expression of SAP97/hDLG isoform (PALM; Fig. 7A), as well as by using an antibody (dSTORM; Fig. 7B) that labels all reported isoforms of SAP97/hDLG, we observed the localization of SAP97/hDLG in different neuronal sub-compartments such as dendritic shafts, morphologically characterized spines and in the proximity of the postsynaptic density (Fig. 6A, B). Additionally, multicolor super-resolution microscopy of the postsynaptic density protein PSD95 and SAP97/hDLG indicated that the majority of condensates are formed peripheral to the postsynaptic density (Fig. 6B). In single synapses where both PSD95 and SAP97/hDLG were colabelled, we found that they overlapped at their periphery and that their center of distribution were within 500 nm (Fig. 6B). We performed dSTORM on unstimulated cultured primary hippocampal neurons of 14 days in vitro (DIV-14) and Neuro-2a cells (Fig. 7B and Fig. 1A). We observed that the molecular fingerprints of nanoscale segregation, namely, $Rc$ and $\Delta G(n_c)$ for endogenously expressed SAP97/hDLG were different between neurons and Neuro-2a cells (Fig. 7C). We then overexpressed Neuro-2a cells with the $I_2$ and $I_3$ isoforms to alter the $I_3$ vs. $I_2$ ratio from its endogenous levels. Ectopic expression of the $I_3$ isoform altered the $I_3$ vs. $I_2$ ratio in Neuro-2a cells towards the endogenous expression as seen in primary hippocampal neurons, while ectopic expression of $I_2$ deviated it farther away (Fig. 7C–E). As indicated previously, the total pool of SAP97/hDLG was studied by super-resolution microscopy. The critical cluster radii of SAP97/hDLG remained similar between Neuro-2a cells expressing $I_2$ and $I_3$ ($R_c$ $I_2$:0.81 ± 0.02, $R_c$ $I_3$:0.88 ± 0.02) isoforms, while it was significantly different between DIV-14 neurons and untransfected Neuro-2a cells ($R_c$ DIV-14:1.20 ± 0.02, $R_c$

Control:1.00 ± 0.03) (Fig. 7D). The nucleation barrier of SAP97/hDLG remained similar between Neuro-2a cells expressing the $I_3$ isoform of SAP97/hDLG and neurons ($\Delta G$ $I_3$:0.87 ± 0.02, $\Delta G$ DIV-14:0.87 ± 0.03), while it was significantly different between neurons and $I_2$ isoforms ($\Delta G$ $I_2$:0.76 ± 0.01, $\Delta G$ DIV-14:0.87 ± 0.03), as well as from the untransfected control ($\Delta G$ Control:1.00 ± 0.02) (Fig. 7E). Our results confirmed that SAP97/hDLG can form isoform-dependent nanoscale condensates in both polarized and unpolarized cells.

## Discussion

Molecular organization of multidomain scaffolding proteins such as SAP97/hDLG is important in subcellular compartments with heterogeneous molecular composition[3,52]. The incorporation of these large scaffolding molecules into multi-protein complexes and their intrinsic ability to form multimers with the same or different family of proteins give them the flexibility to alter their compositionality and their signaling state instantaneously[1,37,53–56]. The last few years have highlighted the significance of fine tuning the molecular self-organization from nano to micron scale at real-time temporal resolution by confined phase transitions at high spatial and temporal precision[57,58]. Several multidisciplinary approaches have been used to evaluate these phase transitions as modulations in local biochemical interaction profiles, conformational changes, post-translational modifications and local exchange kinetics. However, majority of these studies are performed in a controlled environment or as ensemble observations to verify the existence of phase transitions in living cells[35,59,60]. These studies provide valuable insights into the nature of transient organization of molecules in subcellular compartments, but do not provide enough understanding on how it aids the processing and regulation of chemical information at the scale of molecular complexes[32,59–64]. Using SAP97/hDLG, we have evaluated how the local thermodynamic changes contribute to differential sensitivity of this scaffolding protein to [$Ca^{2+}$.CaM]. Since SAP97/hDLG is one of the principal components of several cell adhesion machinery, we evaluated its nanoscale molecular distribution using a combination of direct stochastic optical reconstruction microscopy in total internal reflection mode on molecules close to the adherent membrane of the cell[41,65]. This method allowed us to quantify both the molecular heterogeneity and nanoscale topography of individual SAP97/hDLG molecules in sub-diffraction sized domains on the plasma membrane[34,66]. Our analysis on the clustering properties of SAP97/hDLG clubbed with the morphometric analysis of super resolution images showed first-order phase transitions at multiple scales. We observed that SAP97/hDLG monomers transition into large molecular domains and further into smaller tightly packed molecular domains within these large domains. Our results are consistent with previous observations for membrane-associated scaffolding complexes such as Shank-Homer platforms observed in excitatory synapses[34,67,68].

Many of the interactions of SAP97/hDLG with other molecules have been mapped to specific domains because of its modular architecture[55,69]. Currently there is no experimental evidence to suggest that SAP97 can directly bind $Ca^{2+}$. Over the years, there has been a consensus on the response of SAP97/hDLG to modulation of intracellular $Ca^{2+}$ through its association with [$Ca^{2+}$.CaM][25,70]. The binding of CaM is expected to open the conformation of SAP97/hDLG by exposing its multiple protein domains, thereby facilitating intermolecular interaction and oligomerization[25,26,69]. We evaluated the first-order phase transition of SAP97/hDLG upon altering the intracellular $Ca^{2+}$. Multiple strategies were adopted including elevation of cytosolic $Ca^{2+}$, blocking of CaM binding to SAP97/hDLG and sequestering of intracellular free $Ca^{2+}$. Upon comparing these conditions to

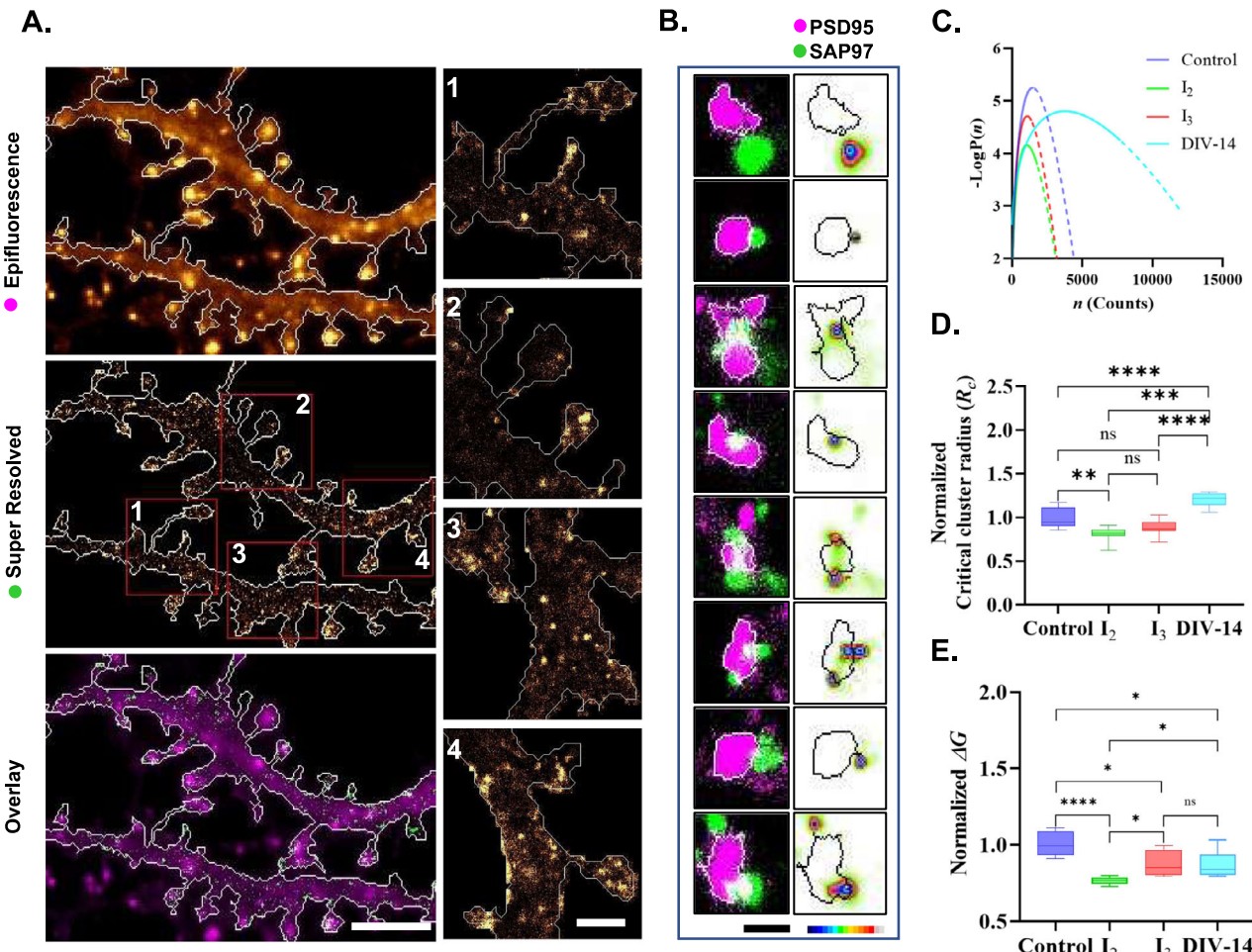

**Fig. 7 Effect of SAP97/hDLG isoform compositionality on nanoscale condensates. A** Ectopic expression of $I_3$-hDLG::mEos in hippocampal pyramidal neurons (DIV-14) shows distinct localization in dendritic shafts and is enriched in morphologically characterized spines (epifluorescence). Super resolution imaging confirms nanoscale condensates (super resolved) formed in dendrites and synaptic compartments (overlay). Scalebar indicates 5 µm. Insets of super resolved images (red regions, super resolved image) are presented to illustrate heterogeneity in nanoscale condensates of SAP97/hDLG in spines and dendritic shafts (1,2,3,4). Scalebar indicates 1.5 µm. **B** Left panel indicates association of SAP97/hDLG condensates with a single postsynaptic density. Postsynaptic density protein PSD95 (Magenta) was identified by semiautomated analysis and its overlap with the nanoscale distribution of endogenous SAP97/hDLG (Green) is presented in white. Scalebar indicates 0.5 µm. Right panel indicates the localization of SAP97/hDLG nanoscale condensates with respect to the nearest identified postsynaptic density (black region). Pseudo color scalebar in all microscopy images represents normalized intensity from maximum to minimum (black to white). **C** Plot of $\Delta G$, derived from the probability density function, with number of molecules per cluster ($n$) indicates the combined molecular fingerprints of critical cluster radius and nucleation barrier between different conditions of SAP97/hDLG modulation in Neuro-2a cells and hippocampal pyramidal neurons. **D** The critical cluster radii of SAP97/hDLG remained similar between Neuro-2a cells ectopically expressing $I_2$ and $I_3$ isoform, while it was significantly different between untransfected neurons and Neuro-2a cells. **E** The nucleation barrier of SAP97/hDLG remained similar between Neuro-2a cells expressing the $I_3$ isoform of SAP97/hDLG and untransfected neurons, while it significantly differed between $I_3$ and $I_2$ isoforms and untransfected Neuro-2a cells. The box represents 25% percentile, median and 75% percentile (Source Data), while the whiskers represents the minimum and maximum values (Source Data). $N = 9$ cells for each condition. Data are cumulated from three experiments. *$p < 0.05$, **$p < 0.01$, ****$p < 0.0001$ (Supplementary Table 6). Source data are provided as Source Data file.

the untreated distribution of SAP97/hDLG, we found a relative increase in the entropy in all cases, resulting in an increased bulk energy component. These results showed that aggregates in untreated cells had the lowest entropy, which increased with any alteration to the system, consistent with the second law of thermodynamics. Though entropy was increased, the discrete alteration of the nucleation barrier and critical radius of clusters across the variable conditions suggested an altered self-organization of clusters. We evaluated how this self-organization could affect entropy by altering its molecular distribution, by comparing a rank-order analysis of both the molecular content and morphological parameters of SAP97/hDLG clusters in each condition with that of unperturbed cells.

Elevation of intracellular $Ca^{2+}$ did not alter the nucleation barrier or critical radius, instead increased the molecular content and size of SAP97/hDLG clusters. Such a molecular rearrangement is energetically favorable for large and stable multi-protein complexes and displays a condition with the least change in entropy, compared to the control conditions. While blocking $Ca^{2+}$-binding protein CaM or on sequestering free $Ca^{2+}$, we observed a significant increase in the nucleation barrier along with increased critical radius. The molecular fingerprints of the nanoscale organization of SAP97/hDLG confirmed that the local changes in chemical information processing could result in modifying entropy by altering the nanoscale aggregation properties in live cells. The smallest molecular domains that can be obtained by

STORM is determined by the localization precision of the microscope[39,71,72]. This results in an inherent limitation to quantify the domains, which are below the resolution limit. Though our results indicate that most of the molecular domains are well above this pointing accuracy, there is a likelihood of under sampling a small fraction of domains, which fall below the resolution limit or do not achieve the desired molecular intensity.

Endogenous SAP97/hDLG modulation resulted in distinct nanoscale fingerprints in multiple conditions. Elevation and sequestering of intracellular $Ca^{2+}$ as well as inhibition of inter-action of $[Ca^{2+}.CaM]$ resulted in differential exchange kinetics for the SAP97/hDLG C-terminal isoforms from the unperturbed condition. In the presence of CaM inhibitor, the phase transition of SAP97/hDLG in Neuro-2a cells increased the molecular content of clusters along with more cytosolic monomer fraction. The reduction in the slope of bulk energy ($b \propto Log(C_{amb}/C_{sat})$) suggested an elevation of the ambient concentration of dispersed phase of SAP97/hDLG. This reduction in bulk energy component correlates to an increase in the entropy of the system. This is consistent with the observation that the mobile fraction of $I_2$ SAP97/hDLG isoform recovered with a shortened halftime of recovery causing an augmented mobility, consistent with the observation of endogenous SAP97/hDLG organization by super-resolution microscopy. In Neuro-2a cells, the mRNA of $I_2$ is almost six-fold more expressed than $I_3$, consistent with the recovery of ectopically expressed $I_2$ isoform of SAP97/hDLG. The $[Ca^{2+}.CaM]$ independent membrane recruitment of $I_2$ would be facilitated better when CaM gets inhibited. While the monomer fraction of such a model would complement the nucleation barrier to be heightened and the super-saturation of molecules to be less favored, attaining it requires spending more energy. In the case of ectopically expressed $I_3$, the mobile fraction increased without any change in the halftime when CaM was inhibited.

Elevation of intracellular $Ca^{2+}$ did not affect the mobile frac-tion of the $I_2$ isoform, while the halftime was significantly increased, this property being influenced by local interactions during the recovery of $I_2$-SAP97/hDLG. Though for $I_2$, this occurred without affecting the nucleation barrier and critical radius of clusters, bulk energy was reduced. This reduction of bulk energy defines an increase in the entropy of the system, but a corresponding change in its nanoscale organization resulted in unchanged nucleation barrier. $I_2$ interacts with CASK, which senses $[Ca^{2+}.CaM]$ through its $Ca^{2+}$/Calmodulin-dependent protein Kinase (CAMK) domain. Since the cytoskeletal interac-tion of CASK is independent of $[Ca^{2+}.CaM]$, the immobilization of $I_2$ is modulated by the activity-dependent interactions of CASK by Mint1/X11α/Lin10 and Liprin α[73,74]. Additionally, CASK is also known to form a tripartite complex with Mint1 and Velis/Lin7 in the rodent brain[75]. This response of $I_2$-SAP97/hDLG would explain a $Ca^{2+}$-mediated decrease in its local recycling rates. Meanwhile, $I_3$-SAP97/hDLG displayed a significant increase in its local recovery rate in the presence of high levels of $Ca^{2+}$. Though the $I_3$ isoform, $I_3$-SAP97/hDLG, is known to associate with both Calmodulin and Protein 4.1, it is unclear whether they could form a tripartite complex potentially dependent on the cytosolic $Ca^{2+}$ level. However, the faster recovery kinetics of $I_3$-SAP97/hDLG at high $Ca^{2+}$ levels indicated a closed conforma-tion, decreasing intermolecular interactions. The mobile fractions of $I_2$ and $I_3$ in this condition were similar to each other. The affinity of $[Ca^{2+}.CaM]$ to CAMK domain is known to be 35-fold higher than intramolecular interactions of CAMK[76], resulting in differential intramolecular alterations and recycling dynamics for SAP97/hDLG isoforms upon elevation of intracellular $Ca^{2+}$.

The differential regulation of SAP97/hDLG isoforms by $Ca^{2+}$ was investigated by chelating intracellular free $Ca^{2+}$. Depletion of free $Ca^{2+}$ in the cells resulted in an elevated nucleation barrier

and critical cluster radius due to a steep decline in the slope of bulk energy, $b$. In comparison to elevated $Ca^{2+}$, the decreased slope of bulk energy indicated a decrease in the ability of SAP97/hDLG to nucleate a tight molecular organization, resulting in dispersed clusters. This reduction in energy could lead to a transient phase with cluster formation and dispersion of this protein. In this case, both the mobile fraction and halftime increased sharply, concurring with the dynamic state of the molecules. Further, to attain super-saturation in this case, the system would need to overcome a higher energy barrier beyond the limit of the experimental system, confirming the availability of $Ca^{2+}$ as a prerequisite for first-order phase transitions that con-trol nanoscale clustering of SAP97/hDLG. In the absence of free $Ca^{2+}$, the system is in a sub-saturated state, complementing the dispersed fraction of SAP97/hDLG compared to a clustered one. FRAP experiments showed a decrease in the mobile fraction of $I_2$: SAP97/hDLG upon chelating $Ca^{2+}$. The rank-order analysis also showed an increase in the recruitment of molecules into an organized domain without similar modifications in the cluster area. In the absence of $Ca^{2+}$, fluorescence recovery of $I_3$ was rescued and reached the levels of $I_2$-SAP97/hDLG, but the half-time of $I_3$-SAP97/hDLG was observed to be significantly higher, confirming an enhanced intermolecular interaction. Association of SAP97/hDLG with $[Ca^{2+}.CaM]$, would increase its propensity to form multimeric protein complexes by unmasking its binding sites. Thus, the compositionality of isoforms would contribute to altering the local kinetics of first-order phase transition, clustering properties and global heterogeneity of this critical scaffolding molecule in response to the flow of chemical information and instantaneous signaling.

$Ca^{2+}$-bound Calmodulin engages with the C-terminal isoforms of SAP97/hDLG differentially. We evaluated this differential dependence by independently inhibiting CaM using w7 and by a stimulation paradigm combining CaM inhibition with elevation of cytosolic $Ca^{2+}$ after activation of SOCE. Upon inhibition of CaM, $I_2$-SAP97/hDLG showed a decrease in its mobile fraction with faster recovery of free molecules. However, elevation of $Ca^{2+}$ along with inhibition of CaM resulted in a slower recovery for $I_2$-SAP97/hDLG, confirming an increase in intermolecular interac-tions independent of $[Ca^{2+}.CaM]$. $I_3$-SAP97/hDLG behaved similar to $I_2$-SAP97/hDLG upon CaM inhibition as well as on co-stimulation by elevating the intracellular $Ca^{2+}$. However, a sig-nificant difference in the rate of recovery was observed between the two isoforms. The increase in the recovery time of $I_3$ can be attributed to the large presence of endogenous $I_2$ isoforms in Neuro-2a cells interacting with ectopically expressed $I_3$-SAP97/hDLG, resulting in intermolecular interactions of both isoforms. Deprivation of $Ca^{2+}$ after inducing SOCE did not alter the mobile fractions of $I_2$ and $I_3$ isoforms from basal conditions. However, both isoforms recovered significantly slower compared to the control. These observations confirmed that $I_2$ and $I_3$ isoforms are significantly affected by the availability of free $Ca^{2+}$, with $I_3$ being preferred via its CaM interaction.

Additionally, we showed that SAP97/hDLG spontaneously nucleate into micron scale condensates in vitro using LLPS at physiologically relevant concentrations. The nucleation of con-densates of $I_3$ isoforms occurred at a lower concentration than that of $I_2$, indicating distinct fingerprints for phase separation allowing differential recruitment of proteins to form co-condensates. In contrast, Calmodulin did not self-nucleate, or phase separate into condensates either exclusively in presence of $Ca^{2+}$ or in the presence of SAP97/hDLG. However, Calmodulin was recruited into SAP97/hDLG condensates when co-incubated with $Ca^{2+}$. The condensates of SAP97/hDLG nucleated Calmo-dulin by LLPS in an isoform-specific manner. Formation of co-condensates of the $I_3$ isoform of SAP97/hDLG occurred at a lower

concentration than that of $I_2$, indicating a $Ca^{2+}$ response window arising from properties of their intrinsically disordered HOOK region. These observations confirmed that SAP97/hDLG can not only form condensates regulated by the HOOK region, but also recruit other proteins into it in response to physiological stimuli such as intracellular elevation of $Ca^{2+}$ or local $Ca^{2+}$ transients. Using Neuro-2a cells and hippocampal pyramidal neurons, we verified that the combination of isoforms in a cell governs its nucleation and saturation kinetics of nanoscale condensates. Though the critical cluster radius was different for all conditions between Neuro-2a cells and neurons, the nucleation barrier remained similar when neurons and Neuro-2a cells had comparable ratio of $I_3$ vs. $I_2$ isoforms. This allowed us to summarize that the nucleation barrier might be a result of intrinsic properties derived from the local compositionality of SAP97/hDLG isoforms, but the critical cluster radius could be influenced by its heterogeneity in interactome, as many interacting proteins of SAP97/hDLG such as synaptic molecules are not present in Neuro-2a cells.

Our observations highlight the intrinsic modulation of SAP97/hDLG in response to alterations in intracellular $Ca^{2+}$, resulting in first-order phase transitions of this scaffolding protein. In vitro, LLPS of SAP97/hDLG into condensates occur by homotypic interactions, while for Calmodulin it is heterotypic and requires the presence of both SAP97/hDLG and $Ca^{2+}$. In vivo, this phase transition is dependent on the nanoscale organization of endogenous SAP97/hDLG, where they are organized in a dispersed phase, a liquid phase, and a gel phase with increased packing density per unit area. These different phases have distinct nanoscale signatures. The phase transition of SAP97/hDLG contributing to its local heterogeneity on the membrane would decide its recycling rates, as verified by the kinetics of the two isoforms, which are distinct only in the intrinsically disordered region at the C-terminus. SAP97/hDLG is a ubiquitous member of the MAGUK family and is widely expressed in most of the cell types. The combination of its isoforms distributed across the different subcellular compartments decide not only the scaffolding of integral membrane proteins, but also serves as a locus for several signaling pathways determining cell fate and function. The differing spatio-temporal properties of SAP97/hDLG isoforms render different molecular fingerprints for each nanodomain, where the compositionality of these domains decides the nature of first-order phase transitions resulting in self-organization of the molecules. Our results show that in spite of similar phase transition dynamics, the organization and exchange kinetics leading to such changes are distinct. Similar studies correlating local thermodynamic variables with organizational deficits would be useful in modeling cell adhesion mechanisms, where flow of chemical signals to transmit information is critical. It remains to be seen whether the thermodynamic regulation of molecular organization would follow classical rules in cell junctions such as synapses. Here, in addition to the flow of chemical information, storage, retrieval and elimination of information translates to memory, potentially sustaining longer than an instantaneous alteration in the chemical behavior of independent molecules.

## Methods

**Animals**. All the procedures in this study were performed according to the rules and guidelines declared in the Compendium of CPCSEA 2018 by the Committee for the Purpose of Control and Supervision of Experimental Animals (CPCSEA), Ministry of Fisheries, Animal Husbandry and Dairying, India. The research protocol (CAF/Ethics/659 and CAF/Ethics/790) was approved by the Institutional Animal Ethics Committee (IAEC) of the Indian Institute of Science.

**Plasmid constructs**. The hDLG constructs tagged with enhanced green fluorescent protein (eGFP) at the N-terminus of the protein sequence were ectopically expressed in Neuro-2a cells for the FRAP experiments. The sequences can be found in the public domain of GenBank data base (accession code: U13896 and U13897 [https://doi.org/10.1073/pnas.91.21.9818])[21].

**Cell culture and transfection**. Neuroblastoma cells (Neuro-2a, ATCC® CCL-131™) were cultured in DMEM supplemented with Glutamax, 1% fetal bovine serum and 1% penicillin–streptomycin at 37 °C. Cells were grown in a 5% $CO_2$ incubator on coverslips (Blue Star, India) of 18 mm diameter and 0.17 mm thickness for 48 h and transfected with Turbofect (Thermo Fisher Scientific, USA)[77]. Before transfection, the cells were incubated in 1 ml serum-free DMEM for 1 h. Plasmid (1 μg) was mixed with Turbofect in 100 μl of DMEM. The cells were then incubated for 5 h at 37 °C, followed by the addition of 2x-FBS supplemented growth media (1 ml) and imaged after 24 h.

**Primary hippocampal culture**. Primary hippocampal neurons were cultured from neonatal Sprague-Dawley rat pups aged postnatal day 0–1 (P0–P1)[78–80]. Hippocampi were dissected out using a stereomicroscope (Olympus, SZ51) in chilled Hibernate-A medium (Gibco-A12475-01) supplemented with B-27-(Gibco-17504044), glutaMAX (Gibco-35050061) and antibiotic-antimycotic (Gibco-240062). The tissue was minced and incubated for 10 min at 37 °C in 0.25% trypsin. The sample was then triturated and centrifuged at 800 x $g$ for 5 min. The resultant pellet was resuspended in Hibernate-A and the supernatant was discarded. This procedure was repeated twice and then the cells were resuspended in the complete Neurobasal-A (Gibco-10888022) media (supplemented with B-27, antibiotic-antimycotic and glutaMAX). The cells were counted and seeded at the density of $1 \times 10^5$ cells/ml on 18 mm (#1.5 optical corrected for 0.17+/−0.01) glass coverslips (coated with poly-D-lysine at a concentration of 100 μg/ml) in a 12-well cell culture plate. Cultures were maintained at 37 °C and 5% $CO_2$ at optimum humidity conditions. Media was complemented at an interval of 7 days. Hippocampal pyramidal neurons were transfected with Lipofectamine 3000 (Invitrogen) $I_3$-hDLG::mEos; where $I_3$-hDLG was fused with mEOS2.3, at 13 days in vitro and imaged within 12–24 h after[78].

**Quantitative-polymerase chain reaction (q-PCR)**. Total ribonucleic acid (RNA) was extracted from the hippocampus and cortex of P0 mice ($n = 9$) and from Neuro-2a cells cultured in T75 flask ($n = 4$) using Trizol™ extraction method (Thermo Fisher Scientific, USA). The primers used for the experiments are mentioned in the Supplementary Table 8. RNA was reconstituted in diethyl pyrocarbonate (DEPC)-treated deionized water. The cDNA was synthesized with high-capacity cDNA reverse transcription kit (Applied Biosystems). The samples were loaded on a 96 well plate and real-time Q-PCR was performed in Quant Studio 7Flex real-time Q-PCR system (Applied Biosystems). The cDNA was tested by PCR (Applied Biosystems, USA) and separated by electrophoresis (Bio-Rad, USA), labeled using Sybr safe (Invitrogen, USA) and visualized on a Gel documentation system (Bio-Rad, USA). The expression of β-2-microglobulin was used as the endogenous control. The analysis was performed with comparative Ct method and the graphs were plotted using Graph pad Prism 6 software. The values were obtained using the equation $N = 2^{−ΔΔCt}$ ($N$ = normalized fold-change, $ΔCt_1$ = difference between the cycle threshold (Ct) of endogenous control and the sample, $ΔΔCt$ = Difference between $ΔCt_{sample1} − ΔCt_{sample2}$. Here, sample1 was taken to be the cortex).

**Immunocytochemistry (ICC)**. Neuro-2a cells were transfected with a reaction mixture of fusion constructs of SAP97/hDLG (1 μg) using Turbofect (2 μl) in 100 μl of DMEM. For dSTORM imaging, Neuro-2a cells and DIV-14 primary hippocampal neurons were fixed using 4% paraformaldehyde and 4% sucrose in PBS for 10 min at 4 °C, quenched with 0.1 M glycine in PBS at room temperature, permeabilized with 0.25% TritonX-100 for 5 min and blocked with 10% BSA in PBS solution for 30 min at room temperature. The cells were incubated with the primary antibodies (SAP97 -Rabbit- APZ-010, Alomone Labs, Jerusalem and PSD95 -Mouse-MA1-046, Invitrogen, USA) diluted in 3% BSA (1:500) for 1 h at room temperature in a humidified chamber. The cells were washed four times (5 min each) with 3% BSA and then incubated with suitable secondary antibody for 45 min followed by 4-time washes of 5 min each. The 12-well plates were wrapped in an aluminum foil and stored in PBS at 4 °C for dSTORM imaging. The antibody specific for SAP97/hDLG was validated using western blot of whole protein extract from rodent brain lysate (Supplementary Fig. 14A, B) and was consistent with the existing reports, while PSD95 antibody has been validated in previous works from the lab[41,80,81]. Imaging was done within 24 h of completion of labeling procedure.

**Subcloning of fusion protein constructs**. eGFP (GeneBank: U13896 [https://doi.org/10.1073/pnas.91.21.9818]) was PCR amplified to introduce 5′-XhoI and 3′-NotI restriction sites and cloned as a XhoI-NotI fragment in the bacterial expression vector pET-28a vector (Merck, USA) to generate pET28a-GFP. The primers used for the experiments are mentioned in the Supplementary Table 8. SH3-$I_2$-GUK and SH3-$I_3$-GUK fragments were PCR amplified (GeneBank: U13896 and U13897 [https://doi.org/10.1073/pnas.91.21.9818] with flanking restriction sites of SalI and HindIII from hDLG constructs[21] and cloned into pET28a-GFP. The positive clones were screened by restriction digestion and confirmed by DNA sequencing

(Eurofins Genomics, Luxembourg). The results are illustrated in supplementary information (Supplementary Fig. 8A–F and Supplementary Table 9).

**Protein purification and expression**. *E. coli* strain BL21(DE3) was transformed with pET-28a-SH3-$I_2$-GUK::GFP, pET-28a-SH3-$I_3$-GUK::GFP and with pET-28a-CaM::mCherry plasmid DNA. The cells were grown at 37 °C to an optical density of 0.8 at 600 nm and protein expression was induced with 0.5 mM isopropyl-β-D-thiogalactopyranoside ((IPTG) Thermo Fisher Scientific, USA) overnight. The next day, the cells were centrifuged at 4000 x *g* for 10 min and the pellets were stored at −80 °C. The bacterial cells were lysed by sonication (Vibra cell, Sonics) with Tris buffer (10 mM Tris-HCl [pH 7.4], 150 mM NaCl, imidazole and protease inhibitor (Roche, Switzerland)). As the proteins were predominantly present in the soluble fraction, the sonicated samples were centrifuged at 30,000 x *g* for 1 h to remove cellular debris. The proteins were purified by Ni$^{2+}$-nitrilotriacetic acid (NTA)-agarose affinity chromatography. The supernatant was passed through Ni$^{2+}$-NTA-agarose resin (Qiagen) and the protein-bound beads were washed with 4 column volumes (~25 ml) of wash buffer containing 10 mM Tris-HCl (pH 7.4), 150 mM NaCl, 50 mM imidazole. 4 ml elution buffer (10 mM Tris-Cl (pH 7.4), 150 mM NaCl and 500 mM imidazole) was added to elute the target proteins. The eluted proteins were concentrated to 500 μl (Vivaspin-Turbo4, 30 kDa) and was ultra-centrifuged at 35,000 x *g* at 4 °C for 30 min to pellet out the protein aggregates.

**Size-exclusion chromatography of fusion proteins**. The Superose®6 increase 10/300GL column was pre-equilibrated with 3 column volumes of saline buffer (10 mM Tris-HCl (pH 7.4), 150 mM NaCl and 2% glycerol) and the protein samples were injected. The whole process was performed in Akta pure® HPLC system using Unicorn7.3 software. The purity of isolated protein fractions of 500 μl were confirmed using SDS-PAGE. The concentration of proteins was measured using NanoDrop (Thermo Scientific). The absorbance measurement at 280 (A280) and the extinction coefficients of the respective proteins were used to calculate their concentration.

**SDS-PAGE and Coomassie Blue staining**. The SDS-PAGE gel was prepared with 10% resolving gel (10% Acrylamide:bis acrylamide, 0.375 M Tris (8.8 pH), 0.1% SDS, 0.1%APS, TEMED) and 4% stacking gel (0.126 M tris (6.8 pH)), 0.1% SDS, 0.1% APS, TEMED). The casting was done with 1.5 mm thickness glass plates (Bio-Rad). 15 μg total protein in 1x Laemmli buffer was loaded per lane and the electrophoresis was carried out in the running buffer (25 mM Tris, 190 mM, Glycine, 0.1% SDS) using the mini-Protean tetra cell system (Bio-Rad). The gels were stained with Coomassie Blue, destained twice with the washing solution (methyl alcohol, acetic acid and distilled water, 4:1:5 ratio) for 1 h, and were imaged using a Gel Doc EZ imager (Bio-Rad).

**Protein structure prediction and modeling**. The amino acid sequences of fusion constructs were uploaded to i-TASSER(Iterative Threading ASSEmbly Refinement) server to obtain the structure predictions with high confidence. The predicted models with the highest c-score of −2.20, −2.00, and −1.00 were chosen for SH3-$I_2$-GUK::GFP, SH3-$I_3$-GUK::GFP and CaM::mCherry, respectively. The reconstruction of the models was performed using UCSF-ChimeraX (Version-1.3). The electrostatic potential map was generated from APBS, an open server from NIH, USA.

**In vitro liquid–liquid phase separation of proteins**. All LLPS experiments were carried out using acid-treated coverslips. The coverslips (Blue star) of 18 mm diameter were kept in concentrated HCL for 12 h and thoroughly washed with deionized water (milli-Q, Millipore) and sterilized. The LLPS solution was prepared from varying concentrations of SEC purified proteins (1 nM, 50 nM, 100 nM, 200 nM, 500 nM, 1 μM, 2 μM) and a molecular crowding agent polyethylene glycol (PEG-4000) (2.5%, 5%, 10%, 20% and 40% (w/v)) in saline buffer (10 mM Tris-HCl (pH 7.4), 150 mM NaCl and 2% glycerol) to determine the phase regime. The reaction mixture was loaded in a Ludin chamber (Life imaging services, Switzerland) for imaging. The images were acquired at an ambient temperature of 25 °C. Calcium chloride was added to the reaction mixture at 2 mM concentration.

**Elevation and modulation of intracellular Ca$^{2+}$ levels**. The pharmacological treatments were performed to alter the intracellular Ca$^{2+}$ levels of Neuro-2a cells. Thapsigargin (Merck, USA), a drug capable of blocking all the four isoforms of the SERCA pump, was applied. Stock solution of 5 mM Thapsigargin was prepared in dimethyl sulphoxide (DMSO) and working concentration of 1 μM in HEPES buffer (extracellular solution/ECS), wherein the cells were incubated for 10 min. Calmodulin (CaM), one of the significant Ca$^{2+}$ sensors in living cells, was blocked by a CaM antagonist w7 (N-(6-Aminohexyl)−5-chloro-1-naphthalene sulfonamide hydrochloride) (Merck, USA). The stock concentration of w7 was prepared in DMSO (100 mM) and working concentration of 25 μM in ECS, wherein the cells were incubated for 30 min. Concentrations of w7 above 50 μM was found to be lethal to cells. 50 mM stock solution of a cell permeable Ca$^{2+}$ chelator named 1, 2-Bis (2-aminophenoxy) ethane-N, N, N′, N′-tetra acetic acid tetrakis (acetoxymethyl ester) (BAPTA-AM: Merck, USA) was prepared in DMSO and its working

concentration of 50 μM in ECS. The cells were incubated for 20 min to chelate the intracellular Ca$^{2+}$. All the treatments of Neuro-2a cells were performed at 37 °C.

**Epifluorescence microscopy**. The Ludin chamber loaded with the sample was mounted on an Axio observer Z1 inverted microscope (Zeiss, Germany) equipped with an Apotome 2 module. The image acquisition was performed using a plan-apochromat 63x/1.4 NA oil immersion objective and excitation/emission filters of GFP and DsRed (Zeiss, Germany). Zen 2 blue edition was used for image acquisition (Zeiss, Germany) software. The images were obtained using an Axiocam 506 mono 6 Mp CCD camera with a frame size of 2752p x 2208p (as czi or tif files) at 8-bit depth. The image processing was done using Fiji (NIH, Bethesda, USA) and MetaMorph (Molecular Devices, San Jose, USA).

**Confocal microscopy**. Neuro-2a cells over-expressing SAP97/hDLG isoforms were imaged on a Zeiss LSM 780 (Carl Zeiss, Thornwood, USA) confocal inverted motorized microscope equipped with a 63x oil objective of refractive index1.5 and NA1.4. The cells were imaged at a sampling of ≈88 nm/pixel. 3D Confocal Z-stacks of total volume (≈2000 μm$^3$) were acquired with single optical sections of thickness 100 nm. ZEN 2011 LSM imaging software was used for acquisition and the image analysis was performed using Metamorph.

**Fluorescence recovery after photobleaching (FRAP)**. FRAP experiments were performed on Neuro-2a cells to study the ectopic expression of alternative splice variants of hDLG fused with eGFP. The cells were cultured on 18 mm coverslips and were loaded on a Ludin Chamber (Life imaging services, Switzerland) supplemented with an extracellular solution suitable for live cell imaging (HEPES buffer-7.5 pH: 10 mM Glucose, 120 mM NaCl, 3 mM KCl, 1 mM MgCl2, 2 mM CaCl2, 10 mM HEPES). The Ludin chamber was mounted on a motorized inverted Olympus microscope (IX83) and was imaged at 37 °C (OKO lab, Italy). GFP excitation was performed using a 488 nm excitation laser in total internal reflection fluorescence (TIRF) mode. Image acquisition was done using a 100x objective (U Apo N 100x Oil immersion TIRF; NA: 1.49) and an EMCCD (Evolve, Photometric, USA). A pulsed 488 nm laser of 100 mW average power was used for bleaching within a region of interest (ROI) of 3 μm diameter at an exposure of 20 ms. After acquiring ten pre-bleach images, the ROI was bleached, and 400 post-bleach images were acquired. The image acquisition and data collection were performed using Metamorph software.

Normalized fluorescence intensity was calculated using the equation, $F_{norm} = (I_{bl} - (I_{postbl}/I_{prebl}) \times 100]/[(I_{postbl}/I_{prebl}) \times 100)$. Here, $F_{norm} =$ normalized fluorescence intensity, $I_{bl} =$ Intensity at the time of bleach pulse, $I_{postbl} =$ Intensity at time $t$ after bleach pulse, $I_{prebl} =$ Intensity at time $t$ before bleach pulse. The recovery curve was fitted using Graph pad Prism 6 software with the formula, $Y = Y_0 + (Plateau - Y_0) \times (1 - \exp(-Kx))$. $Y_0 = Y$ value when $X = 0$, Plateau $= Y$ value at infinite $X$, $K =$ rate constant $(1/x)$, $t½ = \ln(2)/K$

**Single-molecule localization microscopy**. Cells were plated on 18 mm round coverslips and imaged at 37 °C (OKO lab, Italy) in a closed chamber (Ludin Chamber; Life Imaging Services, Switzerland). Stochastic Optical Reconstruction Microscopy (dSTORM) and Photoactivation Localization Microscopy (PALM) was performed at 37 °C in an open chamber (Ludin chamber, Life Imaging Services) mounted on an inverted motorized microscope (IX83 Olympus) equipped with a 100× TIRF, 1.49 NA objective allowing long acquisition by HILO (highly inclined and laminated optical sheet) illumination[40,41,77,82]. MetaMorph software was used to steer the illumination and acquisition. Beads of 100 nm diameter were mounted on coverslips for 10 min (Tetraspeck; Thermo Fisher Scientific, USA) and was used as fiducial markers for lateral drift correction. Immunolabeled cells were imaged in dSTORM buffer, an enzymatic oxygen scavenging solution (catalase, TCEP, glycerin, glucose and glucose oxidase dissolved in Tris-HCl buffer), to induce stochastic activation of sparse subsets of molecules[40,41,82]. The photoconversion of Alexa-647 fluorophores from ensemble to single-molecule density was achieved by illuminating the sample with a 300 mW excitation laser. On attaining an optimal density of 0.01–0.04 molecules/μm$^2$, the illumination laser power was reduced to 150 mW, and the samples were imaged. Five stacks of 4000 frames were obtained at an exposure time of 20 ms, acquiring a total of 20,000 images. For multicolor STORM imaging, we used Alexa 532 (for PSD95) in combination with Alexa-647 as secondary labels, similar to previous studies. PALM was performed using a combination of lasers of 561 nm (150 mW) and 405 nm (minimal power) to attain an optimal density of photoconverted 0.01–0.04 molecules/μm$^{2,41}$.

**Evaluation of first-order phase transitions within nanoscale condensates**. Since the number of molecules ($n$) in a domain is a function of morphology and local distribution of molecules within clusters, it relates to the cluster radius ($R$) in nanometers, as $n = (R/1 nm)^3$. The localization events in a cluster scale to the cube of cluster radius by assuming a uniform distribution inside clusters.[34] Thus, the probability distribution of number of molecules detected inside each cluster resulted in a histogram where the probability of occurrence decreases rapidly towards clusters with more molecules behaving non-uniformly as supramolecular aggregation patterns. The free energy change associated with the clustering of monomers into aggregate has been previously reported as $\Delta G = a(n)^{2/}$

$3 \pm b(n) + c$[34]. The term $\Delta G_{\text{surface}} = an^{2/3}$ represents the molecules' surface energy, superficially localized on the cluster to set an interface between the cluster and the surrounding media. The parameter $a = \sigma(36\pi)^{1/3} \cdot (v_1)^{2/3}$, where $\sigma$ is the energy per unit area and $v_1$ is the average volume occupied by one monomer inside the cluster. Simultaneously, $\Delta G_{\text{bulk}} = \pm bn$ defines the bulk energy term that depends on the SAP97/hDLG ambient monomer concentration ($C_{\text{amb}}$) and the saturation concentration ($C_{\text{sat}}$) at equilibrium with the clustered phase. For a cluster of $n$ molecules, $\Delta G_{\text{bulk}} = \Delta\mu n$ and $\Delta\mu = k_B T \, \text{Log}(C_{\text{amb}}/C_{\text{sat}})$. We then extracted the number of detected SAP97 molecules per nanoscale aggregates, resembling nanodomains. $\Delta G(n)$ was determined from the distribution of cluster sizes using the equation $\Delta G = -k_B T \, \text{Log}(P(n)) + \text{Log A}$ [for $n < nc$][34] (Fig. 1C), where $k_B$ is the Boltzmann constant and T, the temperature (Kelvin). A positive ($C_{\text{amb}} < C_{\text{sat}}$) or negative ($C_{\text{amb}} > C_{\text{sat}}$) sign depicts the cluster's saturation state, i.e., whether the system is in a sub-saturated or super-saturated state. The term "$bn$" was extracted from the data by subtracting "$an^{2/3}$" from $\Delta G$, and the curve was observed to be linear with a negative slope, representing a super-saturated system. The negative bulk energy minimizes a system's free energy, while the positive surface energy tries to maximize it. The balance of these two terms defines the nucleation barrier, above which the cluster formation tends to be spontaneous. $P(n)$ represents the relative frequency distribution of cluster size ($n$) and $n_c$ is the critical cluster size attaining a maximum value of $\Delta G(n_c)$, which is the nucleation barrier. The analysis showed an offset in the curve fit with $-\text{Log}P(n)$, which was contributed by Log A, the $y$-intercept ($c$) of the curve. From the curve fit, the inflection points of the curve gave the critical cluster size, $n_c = (2a/3b)^3$, above which the clusters spontaneously grow at the expense of monomer concentration. The data fit gave the parameter values of $a$, $b$, and $c$ (best fit mean ± s.e.m). On substituting these parameters in the free energy equation, we were able to extract the nucleation barrier $\Delta G(n_c)$ and the critical cluster size $n_c \propto (R_c)^3$. $\Delta G(n_c)$ and critical radius of cluster ($R_c$) were normalized with respect to untreated control dataset.

**Analysis of SAP97/hDLG cluster size using super-resolution microscopy**. A typical single-cell SMLM (single-molecule localization microscopy) experiment acquired with the microscope setup as described previously produced a set of 20,000 images that were analyzed to extract single-molecule position and dynamics. Single-molecule fluorescence events were localized in each image frame and tracked over time by a wavelet segmentation algorithm and simulated annealing, respectively. The algorithm used to derive quantitative data on protein localization and dynamics was run using the PALM-TRACER[40] plugin within the MetaMorph software environment. Under these experimental conditions, single-molecule-based localization method by dSTORM resulted in an accuracy of 21.25 nm for Alexa-647. Fiducial markers of 100 nm diameter (Invitrogen, cat. no. T7279) were used to correct for 2D drift occurring during SMLM acquisition. The clusters were identified based on the minimum average intensity threshold of 3–5 detection events. The total intensity of single molecules was acquired from a binary mask after thresholding the image using the integrated morphometry analysis (IMA) plugin in MetaMorph. Detection of events was thresholded to 42 nm of cluster diameter (21 nm of radius), and the median of total intensities from single-molecule events was taken as the total intensity of a single molecule. From this, the number of molecules in a cluster was obtained by dividing the total intensity of clusters by the median of the total intensity of a single molecule[40,41]. The numbers thus obtained were correlated with numbers of standard scaffolding molecules such as PSD95, Bassoon, AMPA Receptors, Amyloid Precursor Protein and beta secretase molecule BACE1 in primary hippocampal neuronal cultures and APP and beta secretase molecule BACE1 in Neuro-2a cells[40,41,81].

The derivation of the polynomial curve fitting of the normalized probability distribution function of SAP97/hDLG cluster size was followed from a previous study[34,48]. The bin size of the probability distribution for different pharmacological conditions was chosen between $\Delta n = 40$, without the function hitting a noise floor. After a trial-and-error approach with various bin sizes, the range of bins was decided for the different datasets. The cluster numbers in all conditions ranged from 1200 to 1500. The critical cluster size ($Rc$) and nucleation barrier ($\Delta Gc$) for all conditions were calculated and were normalized to the mean of untreated control data.

**Statistical methods**. All the sample sizes were calculated using G*Power (Version 3.1.9.7). For the morphometric data from SMLM, sample sizes were calculated using effect size of 0.1, $\alpha$ error prob of 0.1 and power of 0.9 from 2 to 4 groups. For the FRAP experiments, sample sizes were calculated using effect size of 0.7, $\alpha$ error prob of 0.1 and power of 0.9 from 2 to 4 groups[48,83,84].

FRAP experiments of hDLG-$I_2$ and $I_3$ variants were statistically analyzed and tested with one-way Welch's $t$-test for two sample comparison and Brown–Forsythe and Welch's ANOVA with Dunn's correction for multiple comparison. Quantification of FRAP is shown as normalized average fluorescence intensity and of q-PCR as the fold-change of expression normalized to endogenous control. Values are expressed as mean ± s.e.m.

The critical cluster size ($Rc$) and nucleation barrier ($\Delta Gc$) data from multiple conditions were compared and tested for significance using Brown–Forsythe and Welch's ANOVA with Dunn's correction, with $p < 0.001$ as indicated by ****. Linear regression of $bn$ was performed for all pharmacological conditions to explore the value and slope of the bulk energy component.

The normality of distributions of dSTORM data was checked by D'Agostino & Pearson omnibus normality test and was found to be non-normal. The frequency distributions for constructing the probability density function histograms and the cumulative probability distribution function curves were normalized. K–S test was used to compare the distributions of control, Tg, w7 and BAPTA datasets. For K–S test, the confidence level was set to 0.001 due to the high sample size and $p < 0.001$ was indicated by ****. A previously published method was followed to generate the rank-ordered plots of treated vs. control datasets of average fluorescence intensities and puncta area[48]. The data from all cells were pooled for the control and treated conditions. The exact number of puncta were chosen randomly from both control and treated datasets. These data were rank-ordered from lowest to highest, and the treated data was plotted against the control data, and a linear fit was calculated. The slope of the linear regression gave an arbitrary scaling factor. The accurate scaling factor was acquired using a previously established method[48,85]. Multiple scaling factors were applied to scale down the treated distribution within a range around the arbitrary scaling factor. The scaled values greater than the threshold (the minimum value in the control condition) were included in the analysis (scaled Tg = (Tg/scaling factor) > minimum of control). The datasets were analyzed by K–S test to compare the scaled-treated vs. the control distribution. The scaling factor corresponding to the highest $p$-value was considered as the accurate one. This process was repeated 100 times to obtain the scaling factor ± standard error of the mean.

**Reporting summary**. Further information on research design is available in the Nature Research Reporting Summary linked to this article.

## Data availability

All data associated with this study are in the paper and/or the Supplementary Information. All the mathematical and statistical tools that are used for analysis is available in the public domain. Source data are provided as a Source Data file with this manuscript. Source data are provided with this paper.

## Code availability

The rank-order analysis of SAP97/hDLG clusters was performed using a code written in Jupyter notebook 5.6.0, to derive the scaling of average intensity and area of clusters. Public GitHub repository for the code can be found at [https://github.com/LucisQ/Rank-order.git].

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

## Acknowledgements

The authors thank the Bioimaging facility at IISc, India. We also thank the Central Animal Facility at IISc, Bangalore, India. We thank Prof. A Chishti for the SAP97/hDLG plasmids used in this work. We thank Prof. Aravind Penmatsa and Mr. Smriti Nayak, Molecular Biophysics Unit, IISc for their generous help with the protein purification. We extend our gratitude to Prof. Ibrahim Cisse, MIT, USA and Dr. Arjun Narayanan, Max Planck institute for physics of complex systems, Germany for their comments and suggestions to improve the phase separation analysis of SAP97/hDLG. We thank Prof. Samir K Maji, Dept. of Biosciences and Bioengineering, IIT Bombay for his comments to optimise the in-vitro imaging of SAP97/hDLG condensates. We also extend our gratitude to Ms. Yukti Chopra, Cognitive Neuroscience, SISSA, Italy for her assistance in automating the rank-order analysis. This work was supported by generous grants from Department of Biotechnology (Innovative Young Biotechnologist Award to D.N. and M.J.), Department of Biotechnology Genomics Engineering Taskforce to D.N., D.B.T. Ramalingaswami Fellowship to D.N. and M.J., DBT-IISc Partnership program to D.N. and N.R., IISc-STAR program grant to D.N., Science and Engineering Research Board Early Career Research Award to M.J., Indian Institute of Science (Institute of Excellence Program) and University Grants Commission, India to DN and Tata Trusts, India for the program grant (co-investigator-D.N.). CSIR Senior Research Associateship/Scientists Pool Scheme to N.S. P.R. thanks his doctoral fellowship from Ministry of Education (India) through IISc.

## Author contributions

P.R., M.J., and D.N. designed research. P.R. and D.N. performed all the experiments unless otherwise indicated. P.R. and D.N. performed analysis. A.K., C.B., and J.B.S. developed analysis modules for single-molecule analysis. M.J. prepared the cultures and P.R. and N.R. performed the molecular quantification of isoforms of SAP97/hDLG expressed in mouse brain. P.R., N.R., and N.S. generated fusion constructs, purified and characterized the proteins. P.R., M.J., and D.N. wrote the manuscript. All the authors read, provided critical inputs, and approved the final version of the manuscript.

## Competing interests

The authors declare no competing interests.
