## [Peer Review File · Nature Communications]

Nanoscale Regulation of Ca²⁺ Dependent Phase Transitions and Real-time Dynamics of SAP97/hDLGEditorial Note: Parts of this Peer Review File have been redacted as indicated to maintain the confidentiality of unpublished data.

REVIEWER COMMENTS

Reviewer #1 (Remarks to the Author):

In this manuscript, the authors found that SAP97/hDLG, a key regulator of cell-cell junction organization, formed nanoscale aggregates in Neuro-2a cells in response to Ca²⁺, likely via the HOOK-CaM interaction-induced conformational changes of SAP97/hDLG. SAP97/hDLG have various splicing isoforms, and two of these insertions are located within the intrinsically disordered HOOK region (I2 and I3). Then they observed that SAP97/hDLG isoforms containing I2 and I3 are trafficked differently as they have different sensitivities to the cytoplasmic Ca²⁺ level. So they proposed that the spatial variability of different isoforms of SAP97/hDLG condensates near the membrane is regulated by Ca²⁺. While a model like this sounds interesting, the authors reported no direct evidence to support the conclusion. The experimental paradigm does not seem to fit for the scope of the study. Thus, the findings reported here are too preliminary to warrant publication in NC.

1. The entire paper built on an imaging figure in Fig. 1A. What if the clusters are not formed by phase transition? The single imaging figure without additional experimental corroboration does not support the conclusion drawn. Similarly, fluorescence image of GFP-hDLg in Fig S4A did not present as typical round puncta-like structures formed via phase transition. Since hDLg is easy to be purified, they need to validate the phase transition ability of hDLg using purified proteins in vitro.

2. Related to Q1, the authors believed that “Intrinsically disordered HOOK region of the SAP97/hDLG responds to an elevation of intracellular Ca²⁺ by interacting with Ca²⁺ binding proteins like Calmodulin (CaM). Interaction of the HOOK region of SAP97/hDLG to CaM alters its molecular conformation, destabilizing the interaction between SH3 and GUK domains”. This foundation should be verified with in vitro assays, as this is critical for the whole study.

3. The authors further proposed that perturbation of the intracellular Ca²⁺ levels with chemicals modulated the CaM-SAP97/hDLG interaction, resulted in changes of the phase transition property of SAP97/hDLG. Ca²⁺ changes hundreds of things in cells. One cannot derive any definitive conclusion without rigorous control. The phenomenon the authors observed may arise from indirect consequences. They should validate their findings using in vitro phase transition assay.

4. The cell model that the authors chose is strange. hDLG most likely form clusters in cell-cell contact or polarized cells. So it is strange to image an individual, unpolarized Neuro-2a cells.

Reviewer #2 (Remarks to the Author):

The manuscript deals with the identification of phase order transition in the organization of SAP97 in vivo, which, owing to its size, has to be demonstrated by super resolution spectroscopy. The conclusion drawn was that SAP97 is largely organized on nanoscale level.

Further, the work examines the effect of $[Ca^{2+}]_i$ on the organization and phase transition order of SAP97. Authors dissected various features related to different regions of the molecules and also isoforms.

Evidently, authors have performed several clever experiments and utilized various tools to quantitate the role of Ca^{2+} . And admittedly, some of those experiments are beyond my expertise. I expect that the soundness of the data in experiments such as microscopy would be judged by other reviewer(s).

In my assessment, though the manuscript contains substantial quantity of data, proper communication is lacking. For example, the second result, which has been titled in the manuscript as “Modulation of binding of SAP97/hDLG to Ca^{2+} alters dynamics of first order phase transitions”, wrongly implies that SAP97 binds Ca^{2+} . I am not sure if SAP97 has Ca^{2+} binding motif(s). Ca^{2+} binding is considered highly specific, and binds at a geometry or motif. This needs to be clarified or corrected, since one gets similar impression at multiple instances.

Another statement: “To confirm the CaM independent Ca^{2+} interaction of I2 isoform, the cells were treated.... (para 1 on page 13)” (it implies to me that I2 form binds Ca^{2+}). In my knowledge MAGuKs do not bind Ca^{2+} directly (please correct me if it is wrong!), though being important scaffold protein, they play important role in neuronal plasticity and memory by interacting with a number of proteins in synapse.

CaM binding to SAP97 is Ca^{2+} dependent, and obviously, a perturbation of intracellular Ca^{2+} affects the organization of SAP97 via its interaction with CaM. Thus, a defined dynamical relationship exists between intracellular Ca^{2+} levels and the organization of SAP97.

The manuscript certainly contains interesting results. Authors have cleverly demonstrated roles/dynamics of intrinsically unstructured regions and spliced variants with respect to the change in the $[Ca^{2+}]_i$. However, the fact that it is mediated by CaM should be reflected, even in the subtitle of the manuscript (if it is so).

The language of the manuscript needs to be corrected for basic grammar, typos and proper expression of units.

Reviewer #3 (Remarks to the Author):

Rajeen et al., is describing functional and nanostructural characteristics of a scaffolding platform SAP97/hDLG in Neuro-2a cells. By using a combination of techniques including dSTORM nanoscopy and FRAP they elegantly characterize morphological properties of these hubs at the nanoscale level and link them to organizational phase transitions (of the first order). Notably, they use the number of molecules inside the cluster and pool of free monomers that can be extracted from dSTORM imaging, to evaluate the free energy change (bulk and surface). By perturbing the intracellular calcium, the authors show the role of this ion binding in modulating the phase transition inside the SAP97/hDLG nanoclusters.

Phase transition in molecular aggregates and membrane-less molecular condensates is currently a hot topic across different biological disciplines. Therefore I am confident that the results and the elegant approach presented in this paper will be of interest to the broad scientific community, especially that SAP97/hDLG hub has importance in many cells to cell adhesion systems.

I do not have major remarks, but rather minor suggestions.

-First, it does not transpire from the current manuscript what is the core of the discussed phase transition; structural phase transitions, but what actually could be happening there, e.g. molecules are more packed but is there chemical evidence for that?. A less informed reader would benefit from a more basic introduction to this problem, which would greatly facilitate absorption of the presented methodology, data, and their interpretation.

-The result section is a bit technical. For clarity, maybe the authors would find a way to move some part of the more detailed description of the procedure to the Material and Methods section. This would facilitate understanding of the actual results and their significance.

-Authors showed that blocking the calcium sensor interaction with SEP97/hDLG leads to an increase in the molecules inside the cluster, but only a moderate increase in the cluster area. Could authors discuss whether this, at least in part, could be explained by the resolution limit of their microscope?

This question relates to wherever the changes in the intensity and the area of the cluster are discussed.

-Authors use Kolmogorov-Smirnov test. Why? The data wasn't normally distributed in general?

Title: Nanoscale Regulation of Ca²⁺ Dependent Phase Transitions and Real-time Dynamics of SAP97/hDLG

Response to Reviewers

We are grateful to the editor and the reviewers for evaluating the manuscript positively and for their insightful comments to strengthen the manuscript. We have performed all the additional analyses, experiments to answer the concerns raised by the reviewers. We have added one additional figure to the main manuscript and added five supplementary figures to support the experiments performed for the revised manuscript in addition to alterations in the main text to address the queries raised by the reviewer. We believe that addressing these comments have enhanced the manuscript in its scientific content, technical execution, and its overall quality.

A summary of the additional experiments and modifications for the revised version of the manuscript is listed below. The corresponding alterations in the revised manuscript text are **highlighted in yellow**.

1. In response to the query from reviewer 1 we have added *in vitro* experiments in the revised manuscript to confirm that phase transition of SAP97/hDLG is homotypic and is an intrinsic property of isoforms and that calcium is a pre-requisite for heterotypic condensation of SAP97/hDLG and Calmodulin. This query from reviewer 1 also answers the minor comment from reviewer 3
2. In response to the query from reviewer 1 we have investigated the SAP97/hDLG distribution in Neurons. SAP97/hDLG; both ectopically expressed (PALM) and endogenous (STORM) form nanoscale condensates in polarized cells such as neurons. Majority of SAP97/hDLG condensates in dendritic spines were observed at the periphery of the postsynaptic density. The *in vitro* and *in vivo* data confirm that condensation of SAP97/hDLG is influenced by the combination and/or spatial availability of isoforms.
3. We have also addressed the comments from reviewer 2 and 3 to reduce the conceptual gaps and to improve the readability of paper giving it a broader perspective. We have rectified the typos, simplified the introduction, results, and discussion and edited the language of paper to address the comments.

Reviewer #1 (Remarks to the Author):

In this manuscript, the authors found that SAP97/hDLG, a key regulator of cell-cell junction organization, formed nanoscale aggregates in Neuro-2a cells in response to Ca²⁺, likely via the HOOK-CaM interaction-induced conformational changes of SAP97/hDLG. SAP97/hDLG have various splicing isoforms, and two of these insertions are located within the intrinsically disordered HOOK region (I₂ and I₃). Then they observed that SAP97/hDLG isoforms containing I₂ and I₃ are trafficked differently as they have different sensitivities to the cytoplasmic Ca²⁺ level. So, they proposed that the spatial variability of different isoforms of SAP97/hDLG condensates near the membrane is regulated by Ca²⁺. While a model like this sounds interesting, the authors reported no direct evidence to support the conclusion. The experimental paradigm does not seem to fit for the scope of the study. Thus, the findings reported here are too preliminary to warrant publication in NC.

We are grateful to the reviewer for the encouraging comments. We have answered every query from the reviewer, performed additional *in vitro* and *in vivo* experiments, analysis, and *insilico* observations to address the reviewer's comments. We believe that the suggestions from the reviewer have improved the technical and scientific perspectives of the paper.

Over the last decade, our understanding of the formation, maintenance and elimination of molecular clusters is getting re-evaluated vigorously. In contrast to a controlled environment that we see in solutions, the natural environment in cells offers a different kind of challenge where these molecules transiently interact with more than one molecule at any instant. Such multicomponent interaction offers a very fine scale regulation of membrane associated complexes due to association and dissociation of transmembrane molecules, scaffolds, and other signalling molecules. The research from our group and others have shown that many of these regulatory nanodomains are below the limit of diffraction and are formed at a spatial scale of 10-200 nm transiently at millisecond-minute time scales in cells¹⁻⁵. Recent evidence confirms that molecules involved in these transient domains also participate in liquid-liquid phase separation (LLPS) *in vitro*⁶⁻⁸. Additionally, a long line of research using techniques such as Fluorescent Correlation Spectroscopy, Fluorescence Recovery after photobleaching and Single Molecule Tracking experiments point out that molecules involved in the formation of nano to micron sized domains also have the potential to have altered kinetics of exchange^{1,9,10}. Such differing kinetics arise as a response to cellular environment or modification that happens post translationally, or as a consequence of multicomponent system within a cell. As of now, there are very few studies that offers insights into the nanoscale regulation of many of these clusters in their natural environment. A recent observation identified the potential of advanced microscopy to understand this question directly from cells^{11,12}. To the best of our knowledge, super resolution microscopy is one of the few techniques that can be used to observe the heterogeneity of molecular content of nanoscale structures in a multicomponent system such as a cell. Thus, correlation of super resolution microscopy with FRAP provides a detailed information on these complexes at nanoscale spatial and millisecond temporal precision. Advanced microscopy techniques in combination with novel analysis paradigms provides insights into this regulation, which makes the experimental paradigm suitable and relevant for this study. Additionally, we agree with the reviewer on the need to evaluate if liquid-liquid phase transitions are an intrinsic property of SAP97/hDLG and how these condensates respond to intracellular alteration resulting from an increased elevation in calcium. To address this, we have performed *in vitro* phase transition studies and confirmed that the isoforms of SAP97/hDLG can transition into phase separated condensates and can recruit Calmodulin into them in a calcium dependent manner. These experiments strengthen the results observed using super resolution microscopy and FRAP, allowing us to probe deeper into submicron scale regulation of these clusters, that is not easily accessible by *in vitro* phase separation experiments.

We thank the reviewer for the queries and strongly believe that the revised paper and the additional experiments prove that there is an isoform specific difference in the phase transition of SAP97/hDLG, which contributes to the regulation of its nanoscale condensates.

1. The entire paper built on an imaging figure in Fig. 1A. What if the clusters are not formed by phase transition? The single imaging figure without additional experimental corroboration does not support the conclusion drawn. Similarly, fluorescence image of GFP-hDLg in Fig S4A did not present as typical round puncta-like structures formed via phase transition. Since hDLg is easy to be purified, they need to validate the phase transition ability of hDLg using purified proteins in vitro.

We appreciate this relevant concern of the reviewer. The inferences on the nanoscale regulation of SAP97/hDLG were based on the figures, Figure1, 2 and 3^{11,13,14}. Figure 1A is a representative example of several cells that was used to compare and explain the general framework for acquisition and analysis. This comprehensive analysis pipeline allowed us to reach the conclusion as presented in Figure1B-E in control conditions and Figure 2 between different conditions of modulating the intracellular calcium levels as well as the interaction of calcium with Calmodulin. These methods are comparable to what has been reported previously to explain phase transitions using different forms of microscopy^{11,14}. Figure 3 uses rank order analysis, which provides a holistic insight into morpho-

functional properties of nanoscale condensates, which is in line with previous observations using both microscopy (Spatial regime) ¹³ and electrophysiology (temporal regime) ¹⁵ for factorial scaling of structural or functional properties of these complexes.

Response to reviewer Figure1: ***In vitro* Liquid-Liquid Phase separation of SAP97/hDLG isoforms:** Top and bottom panels indicate the phase separation of C terminal isoforms namely SH3-I₂-GUK::GFP and SH3-I₃-GUK::GFP, respectively in response to increasing concentration of the crowding agent Polyethylene Glycol 4000 (PEG). Concentration of SAP97/hDLG isoforms were maintained at 1μM. Phase transitions were observed above 20% PEG in a saline buffer. Scale bar indicates 5μm.

To the best of our knowledge, purification of full-length SAP97/hDLG has not yet been successful ¹⁶. Our attempts to do the same with full-length SAP97/hDLG, as suggested by the reviewer, met with similar results which resulted either in lack of expression of the full-length protein or bacterial modifications resulting in truncation or addition of multiple mutations (data not shown). To circumvent these challenges, we purified the C-terminal fragment of SAP97/hDLG which contain the SH3-HOOK-GUK region fused to GFP, since the alternate splicing occurred in the intrinsically disordered “HOOK” region. The HOOK region either contained I₂ or I₃ splicing and is referred to as SH3- I₂-GUK::GFP and SH3- I₃-GUK::GFP, respectively. To evaluate the homotypic phase transition of these isoforms, we added increasing concentrations of the crowding agent, Polyethylene Glycol (PEG), to 1μM concentration of SAP97/hDLG in solution. As the concentration of PEG reached 20%, we observed spontaneous phase separation of both SH3-I₂-GUK::GFP and SH3-I₃-GUK::GFP into micron sized condensates (Response to reviewer Figure 1). Next, to examine the dependence of this separation to the concentration of the C-terminal fragment of SAP97/hDLG, we progressively increased the concentration of SAP97/hDLG while keeping the concentration of PEG constant. While LLPS of SH3- I₂-GUK::GFP started to occur at 100nM concentration, SH3- I₃-GUK::GFP partitioned into microscale condensates at low concentrations as 50nM (Response to reviewer figure 2). These results confirm the ability of SAP97/hDLG for homotypic phase transitions with distinct concentration threshold for the evolution of condensates.

Response to reviewer Figure2: **Isoform specific difference in the nucleation of C-terminus of SAP97/hDLG**: Top and bottom panels indicate the phase separation of C terminal isoforms, namely SH3-I₂-GUK::GFP and SH3-I₃-GUK::GFP, respectively in response to their increasing concentration at 20% PEG. SH3-I₂-GUK::GFP nucleation occurs at a concentration higher than SH3-I₃-GUK::GFP, indicating that spontaneous phase transition of I₃ isoform of SAP97/hDLG occurs at a lower concentration than that of I₂ isoform. Scale bar indicates 5µm.

The reviewer is correct that the Figure S4A does not present typical round puncta-like structures formed via phase transition. As of now there is growing evidence of LLPS for several key molecules critical in polarized cell junctions and at neuronal synapses. In comparison to other molecules involved in the control of translation machinery these molecules also form microscale phase separated domains *in vitro*. However, majority of these molecules involved at polarized cell junctions do not form such large clusters in their natural environment^{6,7}. Expression of these molecules in their natural environment result in patterns varying from soluble to punctate structures with diverse size ranges when observed by diffraction limited light microscopy. Interestingly, the size of an average excitatory postsynaptic density falls between 300-600nm, within which these molecules are clustered into domains of 10-200nm in diameter^{1,5-7,11}. The subsynaptic clusters are not resolvable by conventional microscopy paradigms. The results that we present are consistent with this model. The complex multicomponent interactions present within cells may restrict the formation of large micron scale clusters in contrast to a single component system or expressing a truncated protein may allow selective interaction with reduced number or moieties, resulting in an altered macromolecular equilibrium.

2. Related to Q1, the authors believed that “Intrinsically disordered HOOK region of the SAP97/hDLG responds to an elevation of intracellular Ca²⁺ by interacting with Ca²⁺ binding proteins like Calmodulin (CaM). Interaction of the HOOK region of SAP97/hDLG to CaM alters its molecular conformation, destabilizing the interaction between SH3 and GUK domains”. This foundation should be verified with in vitro assays, as this is critical for the whole study.

We thank the reviewer for this query. We have stated in the introduction “*Intrinsically disordered HOOK region of the SAP97/hDLG responds to an elevation of intracellular Ca²⁺ by interacting with Ca²⁺ binding proteins like Calmodulin (CaM). Interaction of the HOOK region of SAP97/hDLG to CaM alters its molecular conformation, destabilizing the interaction between SH3 and GUK domains*” which is a result of several studies that address the behaviour of these domains¹⁶⁻¹⁹. SH3-HOOK-GUK region of SAP97/hDLG and other MAGUKS are present in a compact “C” type conformation facilitating the interaction between these domains²⁰⁻²³. The HOOK region masks the protein-protein interaction sites of SH3 domain²⁴. Previous observations have identified that CaM binds to the HOOK region of MAGUKs with increasing affinity in presence of calcium¹⁷. The foresaid observation of CaM associating with SAP97/hDLG along with the investigation of SH3-HOOK-GUK region of other MAGUKs confirm that the GUK region can interact with the SH3 domain of MAGUKs (SAP97/hDLG). *In vivo*, and *in vitro* studies imply that the binding of CaM or any high affinity ligands in the region between SH3 and GUK can destabilize SAP97/hDLG as well as other MAGUKs as shown by multiple groups over the last 20 years using biochemistry, molecular biology, imaging, structural and *in silico* dynamics^{21,22}. We have not repeated similar *in vitro* assays to avoid duplication of the work generating similar outcome.

[Redacted]

[Redacted]

[Redacted]

[Redacted]

3. The authors further proposed that perturbation of the intracellular Ca^{2+} levels with chemicals modulated the CaM-SAP97/hDLG interaction, resulted in changes of the phase transition property of SAP97/hDLG. Ca^{2+} changes hundreds of things in cells. One cannot derive any definitive conclusion without rigorous control. The phenomenon the authors observed may arise from indirect consequences. They should validate their findings using *in vitro* phase transition assay.

We thank the reviewer for these relevant concerns. To address this, we have evaluated if Calmodulin or SAP97/hDLG form condensates or can recruit other proteins into the condensates formed by LLPS. We were able to show that each SAP97/hDLG isoform results in spontaneous LLPS condensates, albeit at differing concentration. LLPS seen at concentrations as low as 100nM for SH3-I₂-GUK::GFP and SH3-I₃-GUK::GFP is an intrinsic property of C-terminal fragments of SAP97/hDLG. Next, we evaluated whether purified Calmodulin tagged with the monomeric variant of red fluorescent protein Cherry referred to as CaM::mCherry can phase separate alone or at increasing calcium concentrations. CaM::mCherry remained soluble and did not transition into clusters in any of these conditions (Response to reviewer Figure 4). This observation remained consistent upon increasing its concentration at 20% PEG, on increasing the concentration of crowding agent up to 40% or on elevating CaCl_2 in solution up to 2mM. However, in presence of SH3-I₂-GUK::GFP and SH3-I₃-GUK::GFP at 1 μM concentration, though we could observe LLPS for SAP97/hDLG, we did not observe any condensation for CaM::mCherry alone nor did it co-condense with SAP97/hDLG. Nevertheless, in presence of calcium, CaM::mCherry was recruited into condensates of C-terminal fragments of SAP97/hDLG. The result was striking for the conspicuous absence of condensates of one protein exclusive of another (Response to Reviewer Figure 5), confirming that CaM::mCherry was recruited in to LLPS condensates of SAP97/hDLG. Additionally, in line with the response to reviewer question 1, recruitment of CaM::mCherry to SH3-I₂-GUK co-condensates occurred at a higher concentration of CaM (almost double), while keeping the concentration of SH3-I₃-GUK::GFP constant. These experiments confirm that SH3-HOOK-GUK region can recruit CaM into LLPS in a calcium dependent manner, which is a direct consequence of SAP97/hDLG, and calcium bound Calmodulin interaction.

Response to reviewer Figure 4: **Absence of spontaneous phase transitions in Calmodulin:** CaM::mCherry did not phase transition either upon increasing its concentration up to 2 μM or upon increasing the concentration of PEG up to 40% in saline buffer. Scale bar indicates 5 μm .

Response to reviewer Figure 5: **Concentration dependent recruitment of Calmodulin to SAP97/hDLG condensates in presence of calcium:** 1µM SH3-I₂-GUK::GFP and SH3-I₃-GUK::GFP and 2mM CaCl₂ were co-incubated with increasing concentrations of CaM::mCherry from 100nM to 2µM. CaM::mCherry was recruited into SAP97/hDLG condensates when it was 2µM for SH3-I₂-GUK::GFP and 1µM for SH3-I₃-GUK::GFP, respectively. This shows that CaM requires both SAP97/hDLG and calcium for recruitment into liquid liquid phase separated domains of SA97. Additionally, the co-condensation of CaM with I₃ was observed at a lower concentration of CaM, as compared to I₂ isoform. Scale bar indicates 5µm.

4. The cell model that the authors chose is strange. hDLG most likely form clusters in cell-cell contact or polarized cells. So, it is strange to image an individual, unpolarized Neuro-2a cells.

Unpolarized Neuro-2a cells were adopted for this study for the following reasons: 1, To show that the phase transition of SAP97/hDLG is an intrinsic property. 2, In Neuro-2a cells, I₂ isoform of SAP97/hDLG contributes to more than 97% of expressed mRNAs, while in both hippocampal and cortical neurons I₂ and I₃ were present in comparable levels.

One of the hypotheses in our paper is that the local compositionality of isoforms could have a direct consequence on the nanoscale condensation of SAP97/hDLG, as observed by light microscopy. Considering the phase transition of SAP97/hDLG to be dependent on the properties of isoforms, neurons and Neuro-2a cells present different combinations of its C terminal spliced variants. Consistent with previous reports^{26,27}, by ectopic expression of SAP97/hDLG/hDIg isoform (Response to reviewer Figure 6A) as well as using an antibody (Response to reviewer Figure 6B) that labels all reported isoforms of SAP97/hDLG we observed nanoscale condensation of SAP97/hDLG in dendritic shafts, morphologically characterized spines and in the proximity of postsynaptic density (Response to reviewer Figure 6A, B). Additionally, multicolour super resolution microscopy of the postsynaptic density protein PSD95 and SAP97/hDLG indicated that the majority of the condensates are formed peripheral to the postsynaptic density (Response to reviewer Figure 6B). In single synapses where both PSD95 and SAP97/hDLG were colabelled we found that they overlapped at their periphery and that their centre of the distribution was within 500nm (Response to reviewer Figure 6B). Using an antibody that labels all reported isoforms of SAP97/hDLG, we

performed dSTORM on unstimulated neurons and on Neuro-2a cells. We observed that the molecular fingerprints of nanoscale segregation were different between neurons and Neuro-2a cells (Response to reviewer Figure 6C-E). We then overexpressed Neuro-2a cells with the I₃ isoform to improve the I₃ vs I₂ ratio to make it comparable to neurons. In this case, we labelled the total pool of SAP97/hDLG. Interestingly, in Neuro-2a cells ectopically expressing SAP97/hDLG-I₃, characteristics of nanoscale segregation (nucleation barrier) were similar to that in neurons (Response to reviewer Figure 6E). These studies confirm that the phase transition of SAP97/hDLG is influenced by the compositionality of its isoforms (Response to reviewer Figure 6).

Response to reviewer Figure 6: **Nanoscale condensates of SAP97/hDLG in neurons whose molecular fingerprints are influenced by the compositionality of SAP97/hDLG isoforms.** A) Ectopic expression of I₃-hDLG:mEos on hippocampal pyramidal neurons (14 DIV) shows distinct localization in dendritic shafts and is enriched in morphologically characterized spines (epifluorescence) and super resolution imaging confirms nanoscale condensates (Super resolved) formed in dendrites and synaptic compartments (Overlay). Scale bar indicates 5 μm. Insets of super resolved image (red regions, super resolved image) is presented to illustrate heterogeneity in nanoscale condensates of SAP97/hDLG in spines and dendritic shafts (1,2,3,4). Scale bar indicates 1.5 μm. B) Left panel indicates association of SAP97/hDLG condensates with single postsynaptic density. Postsynaptic density protein PSD95 (Magenta) is identified by semiautomated analysis (white regions) and nanoscale distribution of endogenous SAP97/hDLG (Green). Their overlap is presented as white. Scale bar indicate 0.5 μm. Right panel indicates the localization of SAP97/hDLG nanoscale condensates with respect to the nearest identified postsynaptic density (black region). Pseudo colour scalebar in all microscopy images indicated in the figure is represented in normalised scales of intensity maximum to minimum (black to white). C) Plot of ΔG , derived from the probability density function, with number of molecules per cluster (n) indicating the combined molecular fingerprints of critical cluster radius and nucleation barrier between different conditions of SAP97/hDLG modulation in Neuro-2a cells and hippocampal pyramidal neurons. D) The critical cluster radii of SAP97/hDLG remained similar between Neuro-2a cells ectopically expressing I₂ and I₃ isoform, while it was significantly different for untransfected neurons and Neuro-2a cells. E) The nucleation barrier of SAP97/hDLG remained similar between Neuro-2a cells expressing I₃ isoform of SAP97/hDLG and untransfected neurons, while significantly different between

I₃ and I₂ isoforms and untransfected Neuro-2a cells. N for each condition = 8-11 cells, Plots represent mean ± s.e.m. * = p<0.05, ** = p<0.01, **** = p<0.0001.

In summary the suggestions from reviewer allowed us to 1) Confirm that phase transition of SAP97/hDLG is homotypic and is an intrinsic property of isoforms 2) Calcium is a prerequisite for heterotypic condensation of SAP97/hDLG and Calmodulin 3) SAP97/hDLG; both ectopically expressed (PALM) and endogenous (STORM) form nanoscale condensates in polarized cells like neurons. 4) Majority of SAP97/hDLG condensates in dendritic spines were observed at the periphery of the postsynaptic density 5) The condensation of SAP97/hDLG is influenced by the combination and/or spatial availability of isoforms.

Reviewer #2 (Remarks to the Author):

The manuscript deals with the identification of phase order transition in the organization of SAP97/hDLG in vivo, which, owing to its size, has to be demonstrated by super resolution spectroscopy. The conclusion drawn was that SAP97/hDLG is largely organized on nanoscale level. Further, the work examines the effect of [Ca²⁺] on the organization and phase transition order of SAP97/hDLG. Authors dissected various features related to different regions of the molecules and isoforms.

We thank the reviewer for comprehending the crux of the message of the paper, and for appreciating the relevance of the results and scope of the work. We feel that the suggestions from the reviewer have greatly improved the conceptual gaps and readability of the paper, giving it a broader perspective.

Evidently, authors have performed several clever experiments and utilized various tools to quantitate the role of Ca²⁺. And admittedly, some of those experiments are beyond my expertise. I expect that the soundness of the data in experiments such as microscopy would be judged by other reviewer(s). In my assessment, though the manuscript contains substantial quantity of data, proper communication is lacking. For example, the second result, which has been titled in the manuscript as “Modulation of binding of SAP97/hDLG to Ca²⁺ alters dynamics of first order phase transitions”, wrongly implies that SAP97/hDLG binds Ca²⁺. I am not sure if SAP97/hDLG has Ca²⁺ binding motif(s). Ca²⁺ binding is considered highly specific, and binds at a geometry or motif. This needs to be clarified or corrected, since one gets similar impression at multiple instances.

We are grateful to the reviewer for the relevant comments. The reviewer is absolutely correct that SAP97/hDLG to the best of our knowledge **do not** have any motifs to directly bind **Ca²⁺**^{22,24,28,29}. SAP97/hDLG binds to calcium bound Calmodulin and senses the cellular modulation of calcium indirectly. We apologize for this miscommunication and have modified the title as “Modulation of intracellular calcium and Calmodulin alters dynamics of first order phase transitions of SAP97/hDLG”.

Another statement: “To confirm the CaM independent Ca²⁺ interaction of I₂ isoform, the cells were treated.... (para 1 on page 13)” (it implies to me that I₂ form binds Ca²⁺). In my knowledge MAGuKs do not bind Ca²⁺ directly (please correct me if it is wrong!), though being important scaffold protein, they play important role in neuronal plasticity and memory by interacting with a number of proteins in synapse.

The reviewer is correct. Following up the previous point that SAP97/hDLG do not directly bind calcium, we have altered the sentence to: “To confirm the modulation of I₂ isoform to elevated calcium, the cells were treated.... (para 1 on page 13)”. Once again, we apologize for this

miscommunication. Taking the cue from the reviewer comment we have carefully rechecked the paper for similar errors and rectified it.

CaM binding to SAP97/hDLG is Ca²⁺ dependent, and obviously, a perturbation of intracellular Ca²⁺ affects the organization of SAP97/hDLG via its interaction with CaM. Thus, a defined dynamical relationship exists between intracellular Ca²⁺ levels and the organization of SAP97/hDLG.

The reviewer comment is very valid ^{16,17}. Our experiments in response to questions from reviewer 1 gives a direct proof of relationship between intracellular Ca²⁺ levels, CaM and the organization of SAP97/hDLG. However, in a cellular environment, this organization is fine-tuned at a molecular scale. We have emphasized this again in the discussion of the revised manuscript.

The manuscript certainly contains interesting results. Authors have cleverly demonstrated roles/dynamics of intrinsically unstructured regions and spliced variants with respect to the change in the [Ca²⁺]_i. However, the fact that it is mediated by CaM should be reflected, even in the subtitle of the manuscript (if it is so).

We thank the reviewer for this relevant comment. We have emphasized explicitly that alteration in the dynamics of SAP97/hDLG spliced variants with respect to the change in [Ca²⁺]_i is mediated by Calmodulin both in the introduction and discussion section of the revised manuscript.

The language of the manuscript needs to be corrected for basic grammar, typos and proper expression of units.

We thank the reviewer for the comment. The modified manuscript has been proofread for the clarity of scientific work and readability by **three native speakers** working in microscopy and cell biology. We believe that the overall quality of the manuscript has been enhanced by these modifications.

Reviewer #3 (Remarks to the Author):

Rajeev et al., is describing functional and nanostructural characteristics of a scaffolding platform SAP97/hDLG in Neuro-2a cells. By using a combination of techniques including dSTORM nanoscopy and FRAP they elegantly characterize morphological properties of these hubs at the nanoscale level and link them to organizational phase transitions (of the first order). Notably, they use the number of molecules inside the cluster and pool of free monomers that can be extracted from dSTORM imaging, to evaluate the free energy change (bulk and surface). By perturbing the intracellular calcium, the authors show the role of this ion binding in modulating the phase transition inside the SAP97/hDLG nanoclusters.

We are grateful to the reviewer for identifying the core message of the paper, and for appreciating the quality of data and the outcome of results. The comments from the reviewer have helped us to make the paper suited to a broader range of readers with a streamlined flow of results.

Phase transition in molecular aggregates and membrane-less molecular condensates is currently a hot topic across different biological disciplines. Therefore, I am confident that the results and the elegant approach presented in this paper will be of interest to the broad scientific community, especially that SAP97/hDLG hub has importance in many cells to cell adhesion systems.

We thank the reviewer for the kind opinion^{11,14}. We have added additional *in vitro* data and have extended the work to compare molecular fingerprints of this phase transition to varying

compositionality of SAP97/hDLG isoforms. We believe that these experiments complement the overall message of the paper.

I do not have major remarks, but rather minor suggestions.

-First, it does not transpire from the current manuscript what is the core of the discussed phase transition; structural phase transitions, but what actually could be happening there, e.g. molecules are more packed but is there chemical evidence for that?. A less informed reader would benefit from a more basic introduction to this problem, which would greatly facilitate absorption of the presented methodology, data, and their interpretation.

This is a relevant comment from the reviewer which has been addressed in the revised manuscript. The introduction has been edited to include the attention of a diverse audience. The invitro assays that we have performed for phase transitions provide chemical evidence for packing. We request the attention of the reviewer to Response to reviewer figures 1-5 which are also included in the results section of the revised manuscript.

-The result section is a bit technical. For clarity, maybe the authors would find a way to move some part of the more detailed description of the procedure to the Material and Methods section. This would facilitate understanding of the actual results and their significance.

We thank the reviewer for the constructive comments to improve the flow and ease of reading of the paper. Methods pertaining to analysis, have been shifted from the “results” section to “materials and methods”.

-Authors showed that blocking the calcium sensor interaction with SEP97/hDLG leads to an increase in the molecules inside the cluster, but only a moderate increase in the cluster area. Could authors discuss whether this, at least in part, could be explained by the resolution limit of their microscope? This question relates to wherever the changes in the intensity and the area of the cluster are discussed.

The smallest molecular domains that can be obtained by STORM is determined by the localization precision of the microscope^{30,31}. This results in an inherent limitation to quantify the domains which are below the resolution limit. However, our results on the molecular domains are well above this pointing accuracy. The changes in the area were also significant in the analysis as indicated by “p values”. The interesting aspect was that the scaling in area was not directly proportional to the scaling in the molecular content as indicated by the number of SAP97/hDLG molecules detected in these molecular domains. We agree with the reviewer that resolution influences the quantification of microscopy images, but the data have been acquired with the best spatial resolution possible within the scope of the article, which will not change the outcome of the result. We value the reviewer comment and have added further clarification in the discussion section of the revised manuscript.

-Authors use Kolmogorov-Smirnov test. Why? The data wasn't normally distributed in general.

We thank the reviewer for this relevant question on the statistical analysis of the data. Every result was analysed for the normality of the distribution. The data was found to have a non-normal distribution. For this reason, we have chosen Kolmogorov-Smirnov test. We have specified this point in the discussion of the revised manuscript.

References

- 1 Choquet, D., Sainlos, M. & Sibarita, J. B. Advanced imaging and labelling methods to decipher brain cell organization and function. *Nat Rev Neurosci* **22**, 237-255, doi:10.1038/s41583-021-00441-z (2021).
- 2 Rossier, O. *et al.* Integrins $\beta 1$ and $\beta 3$ exhibit distinct dynamic nanoscale organizations inside focal adhesions. *Nature cell biology* **14**, 1057-1067 (2012).
- 3 Nair, D. *et al.* Super-resolution imaging reveals that AMPA receptors inside synapses are dynamically organized in nanodomains regulated by PSD95. *J Neurosci* **33**, 13204-13224, doi:10.1523/JNEUROSCI.2381-12.2013 (2013).
- 4 Kedia, S. *et al.* Alteration in synaptic nanoscale organization dictates amyloidogenic processing in Alzheimer's disease. *iScience* **24**, 101924, doi:https://doi.org/10.1016/j.isci.2020.101924 (2021).
- 5 Padmanabhan, P., Kneynsberg, A. & Gotz, J. Super-resolution microscopy: a closer look at synaptic dysfunction in Alzheimer disease. *Nat Rev Neurosci* **22**, 723-740, doi:10.1038/s41583-021-00531-y (2021).
- 6 Zeng, M. *et al.* Phase Transition in Postsynaptic Densities Underlies Formation of Synaptic Complexes and Synaptic Plasticity. *Cell* **166**, 1163-1175 e1112, doi:10.1016/j.cell.2016.07.008 (2016).
- 7 Zeng, M. *et al.* Reconstituted Postsynaptic Density as a Molecular Platform for Understanding Synapse Formation and Plasticity. *Cell* **174**, 1172-1187 e1116, doi:10.1016/j.cell.2018.06.047 (2018).
- 8 Boeynaems, S. *et al.* Protein Phase Separation: A New Phase in Cell Biology. *Trends Cell Biol* **28**, 420-435, doi:10.1016/j.tcb.2018.02.004 (2018).
- 9 Morris, J. D. & Payne, C. K. Microscopy and Cell Biology: New Methods and New Questions. *Annu Rev Phys Chem* **70**, 199-218, doi:10.1146/annurev-physchem-042018-052527 (2019).
- 10 Sezgin, E. & Schwille, P. Fluorescence techniques to study lipid dynamics. *Cold Spring Harb Perspect Biol* **3**, a009803, doi:10.1101/cshperspect.a009803 (2011).
- 11 Narayanan, A. *et al.* A first order phase transition mechanism underlies protein aggregation in mammalian cells. *Elife* **8**, doi:10.7554/eLife.39695 (2019).
- 12 Heidenreich, M. *et al.* Designer protein assemblies with tunable phase diagrams in living cells. *Nat Chem Biol* **16**, 939-945, doi:10.1038/s41589-020-0576-z (2020).
- 13 Venkatesan, S. *et al.* Differential scaling of synaptic molecules within functional zones of an excitatory synapse during homeostatic plasticity. *ENeuro* **7** (2020).
- 14 Bracha, D. *et al.* Mapping Local and Global Liquid Phase Behavior in Living Cells Using Photo-Oligomerizable Seeds. *Cell* **175**, 1467-1480 e1413, doi:10.1016/j.cell.2018.10.048 (2018).
- 15 Kim, J., Tsien, R. W. & Alger, B. E. An improved test for detecting multiplicative homeostatic synaptic scaling. *PLoS One* **7**, e37364, doi:10.1371/journal.pone.0037364 (2012).
- 16 Paarmann, I., Lye, M. F., Lavie, A. & Konrad, M. Structural requirements for calmodulin binding to membrane-associated guanylate kinase homologs. *Protein Sci* **17**, 1946-1954, doi:10.1110/ps.035550.108 (2008).
- 17 Paarmann, I., Spangenberg, O., Lavie, A. & Konrad, M. Formation of complexes between Ca²⁺-calmodulin and the synapse-associated protein SAP97 requires the SH3 domain-guanylate kinase domain-connecting HOOK region. *J Biol Chem* **277**, 40832-40838, doi:10.1074/jbc.M205618200 (2002).
- 18 Lin, E. I., Jeyifous, O. & Green, W. N. CASK regulates SAP97 conformation and its interactions with AMPA and NMDA receptors. *J Neurosci* **33**, 12067-12076, doi:10.1523/JNEUROSCI.0816-13.2013 (2013).
- 19 Tully, M. D. *et al.* Conformational characterization of synapse-associated protein 97 by nuclear magnetic resonance and small-angle X-ray scattering shows compact and elongated forms. *Biochemistry* **51**, 899-908, doi:10.1021/bi201178v (2012).
- 20 McGee, A. W. & Bredt, D. S. Identification of an intramolecular interaction between the SH3 and guanylate kinase domains of PSD-95. *J Biol Chem* **274**, 17431-17436, doi:10.1074/jbc.274.25.17431 (1999).
- 21 Wu, H. *et al.* Intramolecular interactions regulate SAP97 binding to GKAP. *EMBO J* **19**, 5740-5751, doi:10.1093/emboj/19.21.5740 (2000).
- 22 Zhu, J., Shang, Y. & Zhang, M. Mechanistic basis of MAGUK-organized complexes in synaptic development and signalling. *Nat Rev Neurosci* **17**, 209-223, doi:10.1038/nrn.2016.18 (2016).
- 23 McCann, J. J. *et al.* Supertertiary structure of the synaptic MAGuK scaffold proteins is conserved. *Proc Natl Acad Sci U S A* **109**, 15775-15780, doi:10.1073/pnas.1200254109 (2012).
- 24 Zhu, J., Shang, Y., Chen, J. & Zhang, M. Structure and function of the guanylate kinase-like domain of the MAGUK family scaffold proteins. *Frontiers in Biology* **7**, 379-396, doi:10.1007/s11515-012-1244-9 (2012).
- 25 Nair, D. K. *Insights Into Molecular Mechanisms Regulating the Activity of Multidomain Proteins in Living Cells Using FRET-FLIM: Einblicke in Die Molekularen Mechanismen Zur Regulierung Der Aktivität Von Multidomain-Proteinen in Lebenden Zellen Mit Hilfe Von FRET-FLIM Untersuchungen*, (2008).

- 26 Goodman, L. *et al.* N-terminal SAP97 isoforms differentially regulate synaptic structure and postsynaptic surface pools of AMPA receptors. *Hippocampus* **27**, 668-682, doi:10.1002/hipo.22723 (2017).
- 27 Baddeley, D. *et al.* 4D super-resolution microscopy with conventional fluorophores and single wavelength excitation in optically thick cells and tissues. *PLoS One* **6**, e20645, doi:10.1371/journal.pone.0020645 (2011).
- 28 Muller, B. M. *et al.* Molecular characterization and spatial distribution of SAP97, a novel presynaptic protein homologous to SAP90 and the Drosophila discs-large tumor suppressor protein. *J Neurosci* **15**, 2354-2366 (1995).
- 29 Won, S., Levy, J. M., Nicoll, R. A. & Roche, K. W. MAGUKs: multifaceted synaptic organizers. *Curr Opin Neurobiol* **43**, 94-101, doi:10.1016/j.conb.2017.01.006 (2017).
- 30 Valli, J. *et al.* Seeing beyond the limit: A guide to choosing the right super-resolution microscopy technique. *J Biol Chem* **297**, 100791, doi:10.1016/j.jbc.2021.100791 (2021).
- 31 Mockl, L. & Moerner, W. E. Super-resolution Microscopy with Single Molecules in Biology and Beyond-Essentials, Current Trends, and Future Challenges. *J Am Chem Soc* **142**, 17828-17844, doi:10.1021/jacs.0c08178 (2020).

REVIEWERS' COMMENTS

Reviewer #1 (Remarks to the Author):

In the revised manuscript, the authors provided evidence showing that SH3-HOOK-GUK of SAP97 I2/I3 isoforms underwent phase separation in vitro, and CaM could be recruited into SAP97 condensates to alter their internal dynamics in the presence of Calcium. They further showed that SAP97 form nanoscale condensates in neurons, and the condensation is influenced by the compositionality of SAP97 isoforms. My concerns have been adequately addressed. The revised paper carefully describes an interesting and important study that should be of general interest. I recommend that it is accepted without further revision.

Reviewer #2 (Remarks to the Author):

I went through the revised version, where the authors have taken care of most of my concerns. My major concern was about the notion that indicates Ca²⁺ binding to SAP97. It was a technically and scientifically wrong conception inferring that SAP97 is a Ca²⁺ binding protein. The authors have toned that down in their revision.

However, I see some points still in the manuscript, which I find quite uncomfortable to accept, unless proven scientifically correct (authors claim they are..)

For example,

"Since SAP97/hDLG senses Ca²⁺ through Calmodulin,...."

SAP97 and CaM binding is strictly Ca²⁺ dependent as mapped earlier. An alteration in the intracellular Ca²⁺ level would affect this binding, and hence further downstream functions. This however, does not infer that SAP97 could sense Ca²⁺ (even in the presence of CaM).

Other statements from the manuscript:

"Differential sensitivity of I2 and I3 SAP97/hDLG variants to cytosolic Ca²⁺ through CaM modulate their exchange kinetics.."

"To confirm whether CaM modulates the Ca²⁺ dependent response of SAP97/hDLG,...."

It is safer to avoid such conflicting statements which also indicate the same erroneous notion. CaM has many targets where the association is strictly Ca²⁺ dependent. Some of these targets also undergo structural changes once holo-CaM binds. Will such targets be categorized as calcium-dependent or -sensitive that they sense Ca²⁺ through CaM (as per your method)?

I agree that a pharmacological treatment for elevating the intracellular Ca²⁺ levels in cell culture demonstrates the dynamic changes in the clusters of SAP97. This necessarily does not infer that SAP97 is Ca²⁺ sensitive unless proven in vitro.

While the data are interesting, the interpretation by authors is not acceptable.

Some minor corrections:

Heading in Methods: not clear

Protein expression, purification Expression

Inconsistency: Ca²⁺ (in many places it is represented as calcium). The correct form in Ca²⁺ (calcium ion, and not calcium)

Casual presentation of numbers: 10mM (should be 10 mM)

Reviewer #3 (Remarks to the Author):

After addressing different reviewers' comments, this manuscript has significantly improved. Authors performed new experiments and analyses to carefully address comments, remarks, and questions raised by reviewers. In this new shape, I found that the authors have gathered enough data to support their conclusions, as requested by the reviewers.

The phase transition in cellular condensate (nano, and micro-sized) is becoming a hot topic across different biological disciplines.

This manuscript presents very interesting insights concerning the formation of SAP97/hDLG condensates and their regulation by Ca²⁺ through HOOK-CaM. Notably, phase transitions were observed at multiple scales, from molecular to supramolecular.

New concepts, suit of techniques including SMLM (PALM, dSTORM) and FRAP, and novel quantitative analyses developed by the authors will be of great interest to scientists other than neuroscience disciplines.

Notably, the authors developed a very interesting and original way to extract thermodynamic parameters involved in cellular condensate transitioning from super-resolved images.

Therefore, I am convinced that this manuscript will be of great interest to the broad scientific public interested in the mechanism involved in the self-organization of molecular condensates.

Manuscript ID: NCOMMS-21-37092A

Title: Nanoscale Regulation of Ca²⁺ Dependent Phase Transitions and Real-time Dynamics of SAP97/hDLG

Response to Reviewers

We are grateful to the editor and the reviewers for evaluating the manuscript positively and for their insightful comments to strengthen the manuscript. It has been a delightful experience to address the queries raised by the editor and reviewers, that has greatly improved the final version of the manuscript.

Reviewer #1 (Remarks to the Author):

In the revised manuscript, the authors provided evidence showing that SH3-HOOK-GUK of SAP97 I₂/I₃ isoforms underwent phase separation in vitro, and CaM could be recruited into SAP97 condensates to alter their internal dynamics in the presence of Calcium. They further showed that SAP97 form nanoscale condensates in neurons, and the condensation is influenced by the compositionality of SAP97 isoforms. My concerns have been adequately addressed. The revised paper carefully describes an interesting and important study that should be of general interest. I recommend that it is accepted without further revision.

We are grateful to the reviewer for the encouraging comments and for appreciating our efforts in the revised manuscript to address the reviewer concerns. We believe that the suggestions from the reviewer have significantly improved the technical and scientific perspectives of the manuscript.

Reviewer #2 (Remarks to the Author):

I went through the revised version, where the authors have taken care of most of my concerns. My major concern was about the notion that indicates Ca²⁺ binding to SAP97. It was a technically and scientifically wrong conception inferring that SAP97 is a Ca²⁺ binding protein. The authors have toned that down in their revision. However, I see some points still in the manuscript, which I find quite uncomfortable to accept, unless proven scientifically correct (authors claim they are.) For example, “Since SAP97/hDLG senses Ca²⁺ through Calmodulin,....”

We thank the reviewer for comprehending the crux of the message of the paper, and for appreciating the relevance of the results and scope of the work. We feel that the suggestions from the reviewer have improved the conceptual gaps and readability of the paper, giving it a broader perspective. We are grateful to the reviewer for the relevant comments. The reviewer is correct that SAP97/hDLG to the best of our knowledge **do not** have any motifs to directly bind Ca²⁺¹⁻⁴. SAP97/hDLG binds to Ca²⁺ bound Calmodulin. In the first revision itself, we had taken cognizance of the reviewer comments and requested three of our colleagues to read the paper to check for contextual lack of clarity raised by the reviewer. We apologize for any similar errors in the revised manuscript, which we have addressed thoroughly. 7 additional sentences have been corrected in the currently revised manuscript for addressing the reviewer concern including those given below.

“Since SAP97/hDLG senses Ca²⁺ through Calmodulin, we confirmed the presence of its C-terminal spliced isoforms in the hippocampal and cortical regions of rodent brains as well as in heterologous cell lines” is changed to “Since SAP97/hDLG is reported to associate with [Ca²⁺.CaM], we confirmed

the presence of its C-terminal spliced isoforms in the hippocampal and cortical regions of rodent brains as well as in heterologous cell lines.”

We have also added an additional sentence in the discussion that “Currently there is no experimental evidence to suggest that SAP97 can directly bind Ca²⁺”.

SAP97 and CaM binding is strictly Ca²⁺ dependent as mapped earlier. An alteration in the intracellular Ca²⁺ level would affect this binding, and hence further downstream functions. This, however, does not infer that SAP97 could sense Ca²⁺ (even in the presence of CaM).

We are in resonance with the reviewer comment. Sap97 does not sense or bind Ca²⁺ directly, which has been clarified throughout the manuscript. Any sentence giving a similar message has been corrected accordingly.

Other statements from the manuscript:

“Differential sensitivity of I₂ and I₃ SAP97/hDLG variants to cytosolic Ca²⁺ through CaM modulate their exchange kinetics....”

The statement has been altered as following in the revised manuscript, as per the suggestion from the reviewer. “Differential binding of I₂ and I₃ SAP97/hDLG variants to [Ca²⁺.CaM] modulate their exchange kinetics”

“To confirm whether CaM modulates the Ca²⁺ dependent response of SAP97/hDLG,.....”

The revised manuscript is altered as per the suggestion from the reviewer. The corrected statement reads as follows: “To confirm differential sensitivity of SAP97/hDLG to [Ca²⁺.CaM], we inhibited the...”

It is safer to avoid such conflicting statements which also indicate the same erroneous notion. CaM has many targets where the association is strictly Ca²⁺ dependent. Some of these targets also undergo structural changes once holo-CaM binds. Will such targets be categorized as calcium-dependent or -sensitive that they sense Ca²⁺ through CaM (as per your method)?

We thank the reviewer for the suggestion and apologize for the confusion. The revised manuscript is corrected to clarify that **SAP97/hDLG** does not sense or bind Ca²⁺ directly.

I agree that a pharmacological treatment for elevating the intracellular Ca²⁺ levels in cell culture demonstrates the dynamic changes in the clusters of SAP97. This necessarily does not infer that SAP97 is Ca²⁺ sensitive unless proven in vitro. While the data are interesting, the interpretation by authors is not acceptable.

SAP97/hDLG does not sense or bind Ca²⁺ directly, which has been clarified throughout the manuscript. The interpretation of the manuscript is in resonance with the reviewer’s comments. We apologize for any mistake in conveying the message, which is corrected in the revised manuscript, as per the reviewer suggestion.

Some minor corrections:

Heading in Methods: not clear – The revised manuscript is altered for a better clarity of headings in Methods, as per the reviewer suggestion.

Protein expression, purification Expression- The revised manuscript is altered with the corrected headings, as per the reviewer suggestion.

Inconsistency: Ca²⁺ (in many places it is represented as calcium). The correct form in Ca²⁺ (calcium ion, and not calcium)- The revised manuscript is corrected for this inconsistency, as per the reviewer suggestion.

Casual presentation of numbers: 10mM (should be 10 mM)- The revised manuscript is altered, as per the reviewer suggestion.

Reviewer #3 (Remarks to the Author):

After addressing different reviewers' comments, this manuscript has significantly improved. Authors performed new experiments and analyses to carefully address comments, remarks, and questions raised by reviewers. In this new shape, I found that the authors have gathered enough data to support their conclusions, as requested by the reviewers.

The phase transition in cellular condensate (nano and micro sized) is becoming a hot topic across different biological disciplines.

This manuscript presents very interesting insights concerning the formation of SAP97/hDLG condensates and their regulation by Ca²⁺ through HOOK-CaM. Notably, phase transitions were observed at multiple scales, from molecular to supramolecular.

New concepts, suit of technics including SMLM (PALM, dSTORM) and FRAP, and novel quantitative analyses developed by the authors will be of great interest to scientists other than neuroscience disciplines.

Notably, the authors developed a very interesting and original way to extract thermodynamic parameters involved in cellular condensate transitioning from super-resolved images.

Therefore, I am convinced that this manuscript will be of great interest to the broad scientific public interested in the mechanism involved in the self-organization of molecular condensates.

We are grateful to the reviewer for identifying the core message of the paper and for appreciating the quality of the data and the relevance of the outcome. The comments from the reviewer have helped us to make the paper suitable to a broad range of readers with a streamlined flow of results. We are immensely grateful to the reviewer for the encouraging and constructive remarks.

References

- 1 Muller, B. M. *et al.* Molecular characterization and spatial distribution of SAP97, a novel presynaptic protein homologous to SAP90 and the Drosophila discs-large tumor suppressor protein. *J Neurosci* **15**, 2354-2366 (1995).
- 2 Won, S., Levy, J. M., Nicoll, R. A. & Roche, K. W. MAGUKs: multifaceted synaptic organizers. *Curr Opin Neurobiol* **43**, 94-101, doi:10.1016/j.conb.2017.01.006 (2017).
- 3 Zhu, J., Shang, Y., Chen, J. & Zhang, M. Structure and function of the guanylate kinase-like domain of the MAGUK family scaffold proteins. *Frontiers in Biology* **7**, 379-396, doi:10.1007/s11515-012-1244-9 (2012).
- 4 Zhu, J., Shang, Y. & Zhang, M. Mechanistic basis of MAGUK-organized complexes in synaptic development and signalling. *Nat Rev Neurosci* **17**, 209-223, doi:10.1038/nrn.2016.18 (2016).